# The Apparent Resilience of the Dry Tropical Forests of the Nicaraguan Region of the Central American Dry Corridor to Variations in Climate Over the Last *C.* 1200 Years

**William J. Harvey** [1,*] ⓘ**, Nathan Stansell** [2]**, Sandra Nogué** [3] **and Katherine J. Willis** [1]

1   Long-Term Ecology Laboratory, Department of Zoology, University of Oxford, Oxford OX1 3SZ, UK
2   Department of Geology and Environmental Geosciences, Northern Illinois University, DeKalb, IL 60115, USA
3   School of Geography and Environmental Sciences, University of Southampton, Highfield, Southampton SO17 1BJ, UK
*   Correspondence: william.harvey@zoo.ox.ac.uk

**Abstract:** The Central American Dry Corridor (CADC) is the most densely populated area of the Central American Isthmus and is subject to the greatest variability in precipitation between seasons. The vegetation of this region is composed of Dry Tropical Forests (DTF), which are suggested to be highly susceptible to variations in climate and anthropogenic development. This study examines the vulnerability of past DTF surrounding the Asese peninsula, Nicaragua to climatic and anthropogenic disturbances over the past c. 1200 years. Past vegetation, climate, burning, and animal abundance were reconstructed using proxy analysis of fossil pollen, diatoms, macroscopic charcoal, and *Sporormiella*. Results from this research suggest that DTF have been highly resilient to past climatic and anthropogenic perturbations. Changes in DTF structure and composition appear to be linked to the abundance and intensity of fire. Pre-Columbian anthropogenic impacts on DTF are not detected in the record; however, DTF taxa decline slightly after European contact (1522 C.E.). Overall the DTF for the Nicaraguan region of the CADC were found to be highly resilient to both climatic and anthropogenic disturbances, suggesting that this region will continue to be resilient in the face of future population expansion and climatic variation.

**Keywords:** Dry Tropical Forests; Palynology; Diatoms; Charcoal; *Sporormiella*; Nicaragua; Central American Dry Corridor; Resilience; Palaeolimnology

## 1. Introduction

Significant losses in food production across the Central American Isthmus (Isthmus of Tehuantepec south to the Isthmus of Darien) resulting from a deficit in precipitation at the beginning of the harvest in 2015 rendered an estimated 2.2 million people at risk of moderate or severe food insecurity [1]. This episode is indicative of the vulnerability of the flora, fauna and human population of the Central American Dry Corridor (CADC) to seasonal patterns of rainfall, impacting agrarian practices, food and water security [2,3]. Given this recent episode of apparent vulnerability of the CADC to hydroclimatic changes, the aim of this research was to identify past hydroclimatic changes and assess their impacts upon the dominant vegetation type, Dry Tropical Forests (DTF), for the Central Pacific lowlands of Nicaragua spanning the last c. 1200 years (Figure 1).

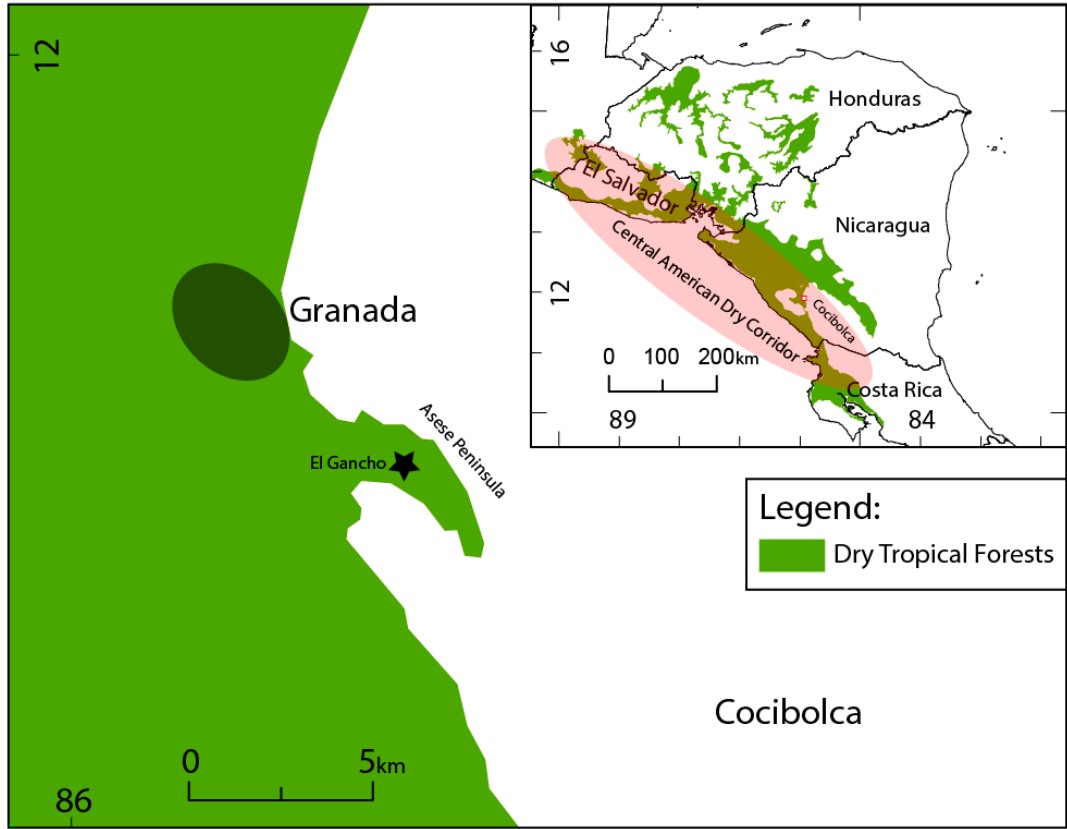

**Figure 1.** Nicaraguan region of the Central American Dry Corridor (ellipsis) and location of Granada, the Asese peninsula and El Gancho.

### 1.1. Climate

Previous research examining the spatial patterns and mechanisms driving precipitation change across the Central American Isthmus over the past 2000 years have focused upon the impacts and timings of hydrological extremes including the Medieval Climate Anomaly (MCA) 950–1250 C.E. and Little Ice Age (LIA) 1400–1850 C.E. [4–20] and the transition between the two [21]. These changes in hydroclimate are reported to be primarily driven by the interactions of three climatic forces: (i) the position of the Inter Tropical Convergence Zone (ITCZ); (ii) the North Atlantic Oscillation; and (iii) the El Niño Southern Oscillation (ENSO) [22]. Warmer sea surface temperatures (SST) in the Caribbean and Gulf of Mexico are inferred for the early part of the MCA [23], and these warmer SST are associated with a more northerly position of the ITCZ as inferred from the Cariaco Basin titanium record [24]. Cooler SST in the Caribbean and Gulf of Mexico were associated with a southward shift in the ITCZ, and a weakening of the summer monsoon during the LIA [5].

Hydroclimate records from Central America spanning the MCA and LIA are reported to be spatially and temporally heterogeneous, with many records lacking sufficient chronological control or resolution to examine this period in detail [22]. Previous work reconstructing past precipitation across the CADC during this time period has been limited with only a single terrestrial $\delta^{18}$O record published from within this region spanning the past 1400 years [16]. This $\delta^{18}$O record indicates comparatively wetter conditions during the MCA and drier conditions during the LIA.

### 1.2. Vegetation

A range of vegetation types are found in the CADC which can be broadly categorised as: (i) dry tropical forests (DTF); (ii) moist tropical forests (MTF); (iii) coniferous tropical forests (CTF); and (iv) mangroves [25]. These formations are primarily distributed according to climate (temperature and precipitation), soil type (e.g., volcanic, alluvial), and elevation (highlands and lowlands). Past research

reconstructing vegetation dynamics within the CADC has been somewhat limited, particularly for DTF and for the last 1200 years. Past anthropogenic influences are typically inferred from agricultural taxa such as *Zea mays*, anthropogenic burning, and evidence from nearby archaeological sites. In comparison, greater forest cover, reduced burning, and an absence of agricultural taxa, appear to coincide with intervals of evidence for reduced population [26]. In contrast, intervals with known archaeological evidence for more people are associated with greater forest clearance (using fire), an abundance of agricultural crops and the increase in abundance of economic or useful taxa [27–29].

In total three sites have been examined from the DTF to date indicating that there have been vegetation responses to known forcing over this interval in time. All three of these studies by Dull [27–29], looked at the vegetation structure and composition of the Ahuachapan region in the CADC of El Salvador, and found that anthropogenic burning between c. 500 B.C.E.–1500 C.E. resulted in a more open grassland environment at the expense of woody taxa. Following this disturbance, Urticales and Mimosoideae were found to have established and were attributed to the population collapse of the region. It is apparent that decreases in arboreal cover are commonly associated with anthropogenic activity, or drier conditions, and more fire; and vice versa for wetter conditions [27].

The resilience of the DTF to climatic and anthropogenic impacts is currently poorly understood [30]. DTF once covered more than 40% of the total area of all tropical forests [31]. They are one of the most threatened habitats globally and are argued to deserve a high priority for conservation [32–34]. It is suggested that either the DTF (i) will be sensitive to climatic change because they are already limited by water and are close to a habitat threshold; or (ii) they will be resilient to changes in rainfall because they are adapted to predictable seasonal drought. Similarly, DTF will either be (i) vulnerable to anthropogenic activities, such as agrarian practices, land conversion and population density because of land use change; or (ii) they will be resilient to anthropogenic factors due to composition adaptation over time [35]. Disturbance, change, and evolution through time can lead to the reorganisation and renewal of the ecosystem [36]. We define resilience by the capacity of a system (biotic or abiotic) to be perturbed while remaining in the same state.

In order to fill these knowledge gaps (detect the impacts of climate and people upon the biota residing within the CADC), this study reconstructed past hydroclimate, biomass burning, DTF dynamics and herbivorous mammal presence, over the last 1200 years (c. 840–2004 C.E). These results were then examined within the context of the archaeological record from the Central Pacific Nicaraguan region. The aim of this study was to (i) identify the past hydroclimatic changes within the Nicaraguan region of CADC; (ii) assess how the DTF responded to identified climatic shifts; and (iii) assess how anthropogenic activities have impacted the DTF of Central Pacific Nicaragua.

### 1.3. Study Area

The CADC is subject to severe and prolonged periods of drought. The dominant type of vegetation in the lowland (<1000 m) areas are DTF [37]. These occur from Chiapas (Mexico) and extend south across the central and lowland areas of Pacific Guatemala through El Salvador, Honduras, Nicaragua, and Costa Rica [38]. DTF occur: (i) in frost-free environments with mean annual temperatures greater than 17 °C; (ii) mean annual rainfall of 250–2000mm; and (iii) overall greater potential of evapotranspiration to precipitation [32]. DTF are subject to a long dry season (5–8 months) and a shorter wet season [2].

The most common taxa found to comprise DTF include: *Bursera simaruba*; *Caesalpinia coraria*; *Calycophyllum candidissimum*; *Guazuma ulmifolia*; *Gyrocarpus americanus*; *Haematoxylon brasiletto*; *Luehea candida*; *Lysiloma kellermani*; and *Phyllostylon brasiliensis* [39].

These formations are typically smaller in stature, lower in biomass and less complex floristically and structurally than tropical rain forests [31]. The canopy is usually closed with no substantial opening in the crown (<20%) reaching a height of approximately 15–20 m [40]. Most plants in the DTF cease their vegetative growth in the dry season and shed their leaves, while other produce fruits and spread their seed [40].

Nicaragua is currently covered by 2500 km$^2$ of DTF, which represents around 2% of the total forest cover [41]. Prior to European contact, savannas were created within the dry tropical forests by inhabiting human populations, especially along the Pacific coast [42,43]. The DTF were more intensively anthropogenically managed compared to the MTF due to the abundance of more productive soils for crop rotation, also known as milpa [31,44]); however, management practices were inefficient in clearing land, allowing for immediate regeneration after several years of cultivation [44].

The Pre-Columbian population inhabiting Nicaragua (before 1522 C.E.) is unknown; however, several attempts to estimate total population have been made, with upper estimates of c. 800,000 [45] to 1,000,000 [46–48] and lower estimates of 100,000 [49]. With European contact and the Spanish conquest of Nicaragua, indigenous populations were reported to have been significantly reduced from 500,000 to 200,000 by 1535 C.E. [50]. Latter conquests and occupation of the Meseta Central of Costa Rica (in 1570 C.E.) reported further depopulation of up to 90% down to c. 61,000 though warfare, slavery and introduction of epidemic diseases [45,51,52]. By the end of the 16th Century there were only around 500 Spanish colonists living in Nicaragua [53]. Populations are suggested to have only started to increase again by c. 1800 C.E (c. 83,000) [51] and reach upper estimates of Pre-Columbian contact levels (c. 1,000,000) by c. 1950 [54]. The current population is around six times (c. 6,272,133) that of the maximum estimates for pre-contact levels [55].

Small-scale deforestation began in 1522 C.E. with the arrival of the Spanish [42,56–58], and continues through present-day. Pasture conversion for cattle was the main cause of deforestation in Nicaragua, which is thought to have impacted DTF more extensively than MTF [32,59,60]. After the Central American Act of Independence in 1821 C.E. landless peasants began to clear land in the DTF [58,60]. DTF were popular settlement areas because land was easier to clear, soil fertility was higher, diseases were rare, and high-quality timbers could be found within these forests [31,59]. In the early 1900s timber extraction, primarily from the DTF, increased for export [57,61]. Targeted species included: *Swietenia humilis, Cedrela odorata, Pachira quinata, Dalbergia retusa,* and *Guaiacum sanctum* [61]. The major causes of DTF loss in recent years are due to agricultural expansion for coffee plantations, crop fields and further ranches [62]. Only fragments of DTF remain today [41].

## 1.4. El Gancho

El Gancho (11.906°N, 85.918°W, 44 m above sea level) is a small, shallow (1.1 m), closed basin lake, situated on top of the Asese peninsula. The Asese peninsula protrudes into the northwest quadrant of Cocibolca (Lake Nicaragua) next to the city of Granada founded in 1524 C.E. (Figure 1). El Gancho is thought to have been formed sometime after 140–345 C.E. when the Asese peninsula was created as the result of a c. 57 km$^2$ debris avalanche originating from the northeast flank of the Mombacho Volcano [16,63,64]. Sedimentological analysis suggests that sediments have continuously accumulated in El Gancho since its formation [16,64]. Total annual rainfall is 1298 mm and mostly falls between May–October [65]. Observations in the field between 2014–2017 C.E. capture variations in lake level derived from precipitation and evaporation: (i) before, (ii) during, and (iii) after a two-year ENSO event (Figure 2).

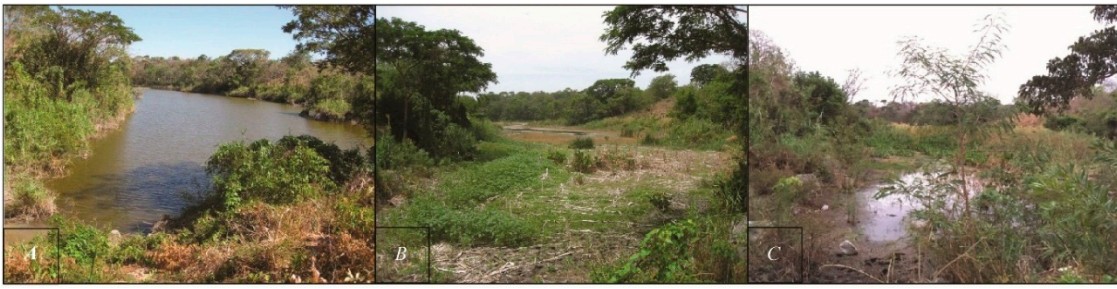

**Figure 2.** El Gancho lake level and vegetation cover. (**A**) Pre-El Nino Southern Oscillation: March 2014; (**B**) El Nino Southern Oscillation: May 2015; (**C**) Post- El Nino Southern Oscillation: March2017.

The vegetation assemblage currently on the Asese peninsula is comprised of taxa belonging to the DTF [25,66,67]. The vegetation is comprised of a higher canopy, primarily deciduous trees: e.g., *Bursera simaruba*, *Gliricidia sepium*, *Diospyros acapulcensis*, *Cochlospermum vitifolium*, and *Myriocarpa bifurca*; and an understory which is typically comprised of sparse and scattered evergreen or deciduous shrubs, and grass tufts, e.g., *Amaranthus spinosus*, Compositae, and Poaceae [33,39,66–69]. The understory is typically burnt by landowners to cleanse unwanted weeds and ticks and to prepare the land for agriculture or ranching [69].

The area surrounding the Asese peninsula, was once DTF but now predominantly agricultural comprising cattle ranching and the growing of corn (*Zea mays*), beans (Leguminosae) and rice (Oryza). Most of this agriculture is grown for export [70].

There are few wild animals left in this area due to extensive land management (e.g., cattle ranching and tourism); however, the mantled howler (*Alouatta palliata*), domesticated dogs (*Canis*), various species of iguana (*Iguana*) and rabbit (*Sylvilagus*) can be found on the Asese peninsula and surrounding islands (Las Isletas). The Asese peninsula is sparsely populated with only one or two dwellings; however, the city of Granada (less than 10 km away), is home to c. 124,000 people [71].

## 2. Methods

### 2.1. Field and Sampling Techniques

In June 2004 a bathymetric survey of El Gancho was conducted using a handheld sonar to identify the deepest part of the lake (1.1 m) from which a 277 cm composite core, with overlapping sections, was retrieved using a Livingstone piston corer [72]. The upper ~168 cm of the core contains massive, very dark brown to black, organic-rich sediments with no notable changes in the stratigraphy [16]. The top 34 cm of the core were extruded in the field at 0.5 cm intervals to ensure sediments did not mix. Forty-two subsamples (1 g wet weight) were extracted at approximately 4 cm intervals down through the upper 168 cm of this sequence for biological proxy analysis: macro-charcoal, pollen, and dung fungal spores (*Sporormiella*); and twenty-two subsamples (0.1 g wet weight) for diatom analysis at an interval of 8 cm (3.5–14.5 range). The biological proxies in the lower 109 cm of the core were found to be degraded and were therefore not analysed further.

### 2.2. Chronology

An age-depth model was constructed using five calibrated radiocarbon dates conducted on charcoal fragments published in Stansell et al. [16]. The radiocarbon dates were (i) recalibrated against the IntCal13 radiocarbon dataset [73]; and (ii) converted to calendar ages and modelled using OxCal 4.3 (Table 1). The Bayesian model applied used a Poisson distribution and made no assumptions of the priors. The Poisson parameter $k_0$ was taken as 1 depth$^{-1}$, the interpolation rate as 0.7 depth$^{-1}$ and the $\log_{10}(k/k_0)$ as U(−2,2). The sedimentation rate per year was calculated from the results of the age-depth model and used in the calculation of pollen concentrations and charcoal influx.

**Table 1.** Measured, calibrated and modelled $^{14}$C ages from El Gancho.

| Lab # | Measured Age ($^{14}$C cal B.P.) | | Depth (cm) | 2σ Calibrated Age Range (A.D.) | | 2σ Median Calibrated Age | 2σ OxCal Modelled Age (A.D.) | | Model Agreement Index | Median Modelled Age |
|---|---|---|---|---|---|---|---|---|---|---|
| UCI-19881 | 630 | ± 35 | 82.25 | 1286 | 1400 | 1343 | 1288 | 1405 | 99.1 | 1346.5 |
| UCI-19882 | 860 | ± 35 | 117.25 | 1046 | 1260 | 1153 | 1053 | 1253 | 105 | 1153 |
| UCI-19883 | 1100 | ± 30 | 162.25 | 887 | 1013 | 950 | 775 | 1012 | 90.7 | 893.5 |
| UCI-22766 | 1640 | ± 40 | 212.75 | 266 | 538 | 402 | 333 | 530 | 110.6 | 431.5 |
| UCI-22767 | 1770 | ± 30 | 226.25 | 123 | 345 | 234 | 186 | 385 | 103 | 285.5 |

### 2.3. Diatom Analysis

Fossil diatoms were used to infer past changes in lake level and area. They were cleaned and prepared using standard procedures as outlined in Berglund [74] and followed the Oxford Long-Term Ecology Laboratory diatom protocol [75]. Samples were mounted using Naphrax resin, chosen for its high refractive index, 1.73 [76]. Counting and identification of diatoms were conducted at 400x and 1000x magnification. A total of 300 frustules per level were counted and morphologically identified (see supplementary materials Table S1) using published keys [77–83]. Annual diatom concentrations were calculated using the evaporative tray method and results from the age-depth model [74]. The diatom samples were matched to the sampling resolution of the palynological data through linear interpolation for ease of comparative analysis.

### 2.4. Fossil Pollen and Dung-Spore Analysis

Fossil pollen was used to reconstruct the composition of past DTF vegetation. Fossil pollen extraction and preparation followed standard palynological procedures applying the Oxford Long-Term Ecology Laboratory protocol [84]. Silicone oil was used as the mounting agent to allow for rotation of grains, easing identification. Samples were spiked with known concentrations of *Lycopodium* spores (batch No. 212761 or 3862) to calculate pollen accumulation rates [85]. Counting and identification of pollen grains were conducted at 400x and 1000x magnification. For each level, a minimum of 300 terrestrial pollen grains were counted. Morphological identification (see supplementary materials Table S2) was achieved using (i) pollen databases [86–88]; (ii) published plates: [89,90]; and (iii) botanical reference materials from the OxLEL reference collections and specimens collected in the field (see supplementary materials Figure S1). *Amaranthus spinosus* was removed from the total palynological counts due to overrepresentation within the assemblage [91]. Total abundance of *Amaranthus spinosus* averaged to 73% of total pollen abundance with a maximum abundance of 90%. Overrepresentation has been attributed to growth within the catchment area and also because species in the Amaranthaceae family are known for their high production of pollen and seed [92]. The abundance of coprophilous dung fungal spores (*Sporormiella*) were used to indicate herbivorous mammal abundance. *Sporormiella* spores were counted and morphologically identified (from their sigmoid aperture) on the same slides [93,94].

### 2.5. Macroscopic Charcoal Analysis

Macroscopic fossil charcoal (>150 μm) was used as a proxy to indicate past occurrence of local fires, representing burning within a 10 km radius of the lake basin [95–100]. Fragments were separated from the sample using a sieve [101]. All charcoal over 150 μm was counted at 10x magnification.

### 2.6. Data Handling

Pollen and diatom counts were converted to percentages while *Sporormiella* and macroscopic charcoal were converted to influx using the exotic marker (*Lycopodium*) and the sedimentation rate [95,102–105]. To identify discrete zones in the palynological assemblage, constrained hierarchical clustering upon the palynological data set was conducted following the broken stick model [106]. Before performing all ordinations analyses a square-root transform was applied to the percentage data to normalise the distribution. This method of transformation was chosen because it can be applied to data sets containing zero values [107]. To check which ordination method was most appropriate for the palynological and diatom data sets, detrended correspondence analysis was conducted and the number of standard deviations away from the mean was assessed [107]. Principal component analysis (PCA) was selected and used to infer similarities between samples and taxa applying a singular value decomposition of the centred but not scaled data matrix. Canonical correspondence analysis (CCA) conducted to quantify the relationship between environmental variables and the palynological and diatom assemblage data. $\delta^{18}O$ data from Stansell et al. [16] was matched through interpolation to the

resolution of the palynological and diatom data sets for the CCA. Ellipses representing the discreet Zones were calculated using the parameterization (cos(theta + d/2), cos(theta − d/2)), where cos(d) is the correlation of the parameters was applied [108]. These were conducted at a confidence level of 95%. These analyses and the presentation of data was performed using R [109], applying packages Vegan [110] and Rioja [111].

## 3. Results

### 3.1. Chronology and Resolution

Re-calibration and modelling of the five radiocarbon dates published by Stansell et al. [16] indicated that the recovered composite core encompasses the past ~1700 years in continuous sedimentation (Figure 3). Distance between subsamples for palynological analysis represent an average resolution of *c.* 30 years. The age-depth model presents good overall agreement (index = 102.9) with an average age range of 190 years surrounding the median modelled age. The model agreement index varied between 99.1 and 110.6 (Table 1).

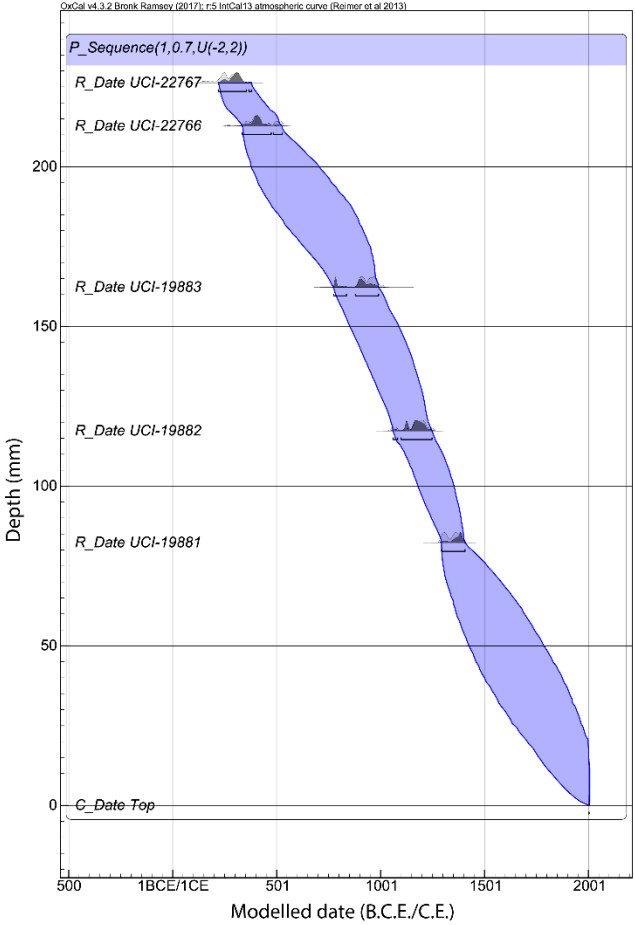

**Figure 3.** Age-depth model for El Gancho.

### 3.2. Palaeoecological Trends: Diatoms

The diatom assemblage is well represented by a dynamic assemblage of facultatively planktonic, planktonic, benthic and epiphytic taxa (Figure 4). Changes within the diatom assemblage broadly reflect the palynologically defined Zones which have been superimposed.

Zone 1 (168–117 cm, 5 samples, 850–1180 C.E.). Benthic (32.2–60.7%) and ephyphitic (18.5–2.3%) taxa dominate this first zone (*Mastogloia smithii, Nitzschia frustulum, Navicula longicephala, Nitzschia amphibia,*

*Encyonema silesiacum* and *Anomoeoneis vitrea*), but decrease in abundance after 1030 C.E. (90.8–48.1%). *Cyclotella meneghiniana* rises from 1030–1150 C.E. (6.1–20.3%) as does *Pseudostaurosira brevistriata* (2.4–23.4%). Total frustule abundance is lowest in this zone (6.6–2.5 x$10^6$ frustules 1 g$^3$ dry sediment).

Zone 2 (107.5–58 cm, 10 samples, 1180–1550 C.E.). Benthic and ephyphitic taxa diminish. *Pseudostaurosira brevistriata* becomes the dominant taxa between 1150–1230 C.E. (23.4–59.9%). After 1340 C.E. *Aulacoseira ambigua* (47.3%) succeeds *Pseudostaurosira brevistriata* (27.4%) in abundance, dominating the assemblage until the end of the zone (41.2%). *Cyclotella meneghiniana* rises between 1450–1530 C.E. (12.6–32.3%). Frustule counts gradually increase (2.4–4 x$10^6$ frustules 1 g$^3$ dry sediment).

Zone 3 (55.5–3.5 cm, 7 samples, 1550–2004 C.E.). From 1600 C.E. *Pseudostaurosira brevistriata* (45.8%) sucedes *Aulacoseira ambigua* (29%). All other taxa diminish except for *Nitzschia frustulum* which increases from 1730–1970 C.E. (3.1–11.8%). *Cyclotella meneghiniana* remains in high abundance until 1600 C.E. (12.1%). Diatom frustules increase from 1570–1600 C.E. (8,863,231–45,552,185 frustules 1 g$^3$ dry sediment) and 1810–1930 C.E. (37,353,262–95,983,203 frustules 1 g$^3$ dry sediment), and decrease in number from 1700–1810 C.E. (63,663,242–37,353,262 frustules 1 g$^3$ dry sediment).

Ordination of the diatom assemblage data denotes clear taxonomic associations with each of the superimposed palynological Zones (Figure 5). The PCA explains 40.2% of the assemblage variance along Axis 1 and 23.4% along Axis 2 (Figure 5a). Zone 1 is predominantly associated with benthic and ephyphitic taxa (e.g., *Mastogloia smithii* and *Nitzschia frustulum*, *Navicula longicephala*, *Nitzschia amphibia*, *Encyonema silesiacum* and *Anomoeoneis vitrea*); Zone 2 is most associated with *Aulacoseira* and *Navicula* taxa; while Zone 3 defined by *Pseudostaurosira brevistriata* and *Pinnularia viridis*. Results from the CCA show a clear significant association with $\delta^{18}$O along the first axis. More positive $\delta^{18}$O are most associated with Zones 2 and 3 (Figure 5b). The $\delta^{18}$O record presented in Stansell et al. (2013) was interpreted to represent drier conditions with more positive $\delta^{18}$O values and wetter conditions with more negative $\delta^{18}$O values.

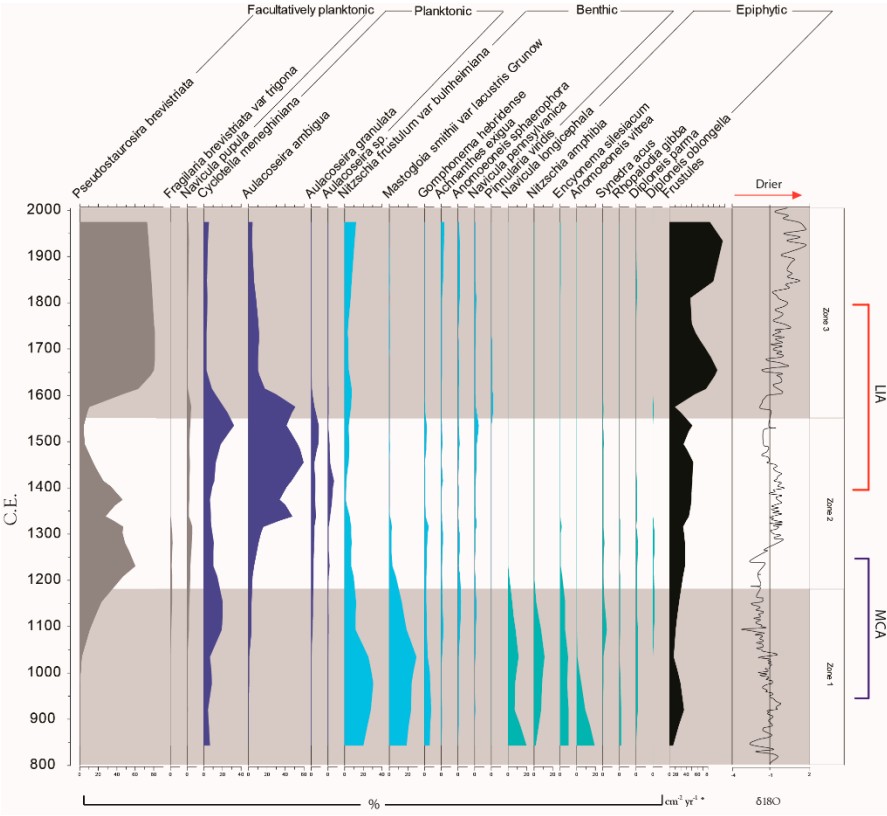

**Figure 4.** Diatom assemblage reconstructed from El Gancho with $\delta^{18}$O data from Stansell et al. [16].

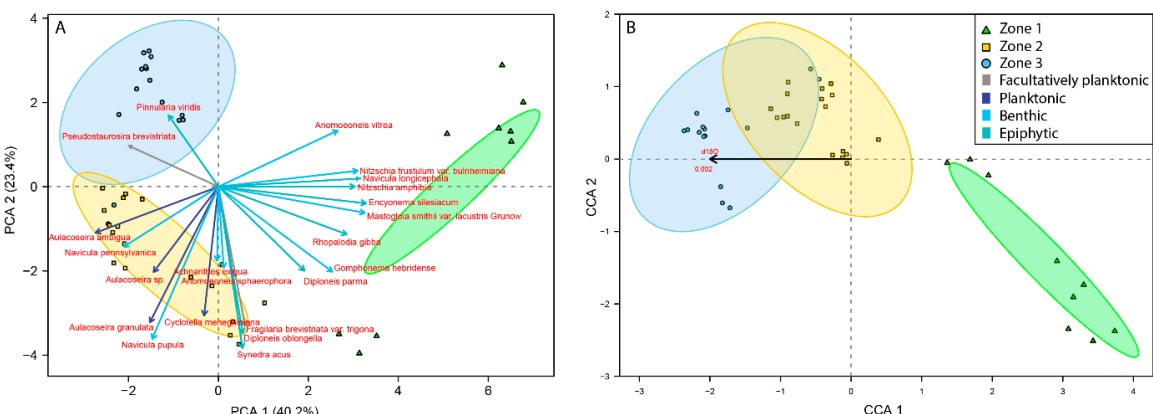

**Figure 5.** Principal component analysis of the diatom assemblage (**A**) Canonical correspondence analysis of the diatom assemblage (**B**) Zones derived from the broken stick model conducted on the palynological assemblage represented by ellipses at a confidence of 95%. Zone 1 = green triangles; Zone 2 = yellow squares; Zone 3 = blue circles.

### 3.3. Palaeoecological Trends: Pollen, Sporormiella and Macroscopic Charcoal

Zonation of the palynological sequence indicates that three numerically derived zones can be recognised between 840–2004 C.E. (Figures 6 and 7). DTF taxa (e.g., *Bursera simaruba*, *Brosimum*, *Protium and Terminalia*) are well represented throughout the entire 1200 years analysed, comprising 48–95% of the total palynological assemblage; however, the DTF community composition varies through time (Figures 6 and 7).

Zone 1 (168–117 cm, 9 samples, 850–1180 C.E.). *Bursera simaruba* (27.1–34.5%), is abundant in the first zone of this sequence, together with Leguminosae (11.5–15.5%), *Protium* (10–12.1%) and *Mimosa* (2.9–8.6%) from the DTF taxa. Taxa comprising the herbaceous understory (Poaceae, Compositeae, Cyperaceae and *Spananthe paniculata*) are abundant (33.3–17.3%) particularly at 890 C.E. (36.7%), 1000 C.E. (39.3%) and between 1090–1120 C.E. (36.1–38.2%). Non-DTF arboreal taxa are represented in low abundance by *Pinus* (1.4–4.4%), *Alchornea* (0–1.4%) and *Roupala montana* (0–0.9%). *Sporormiella* influx is high at 980 C.E. (2.6 spores cm$^{-2}$yr$^{-1}$) between 1030–1090 C.E. (4.3–3.3 spores cm$^{-2}$yr$^{-1}$) and at 1150 C.E. (4.5 spores cm$^{-2}$yr$^{-1}$). Macroscopic charcoal influx is high (10.3–38.1 particles cm$^{-2}$yr$^{-1}$), peaking between 920–980 C.E. (33.2–29.8 particles cm$^{-2}$yr$^{-1}$) and at 1090 C.E. (38.1 particles cm$^{-2}$yr$^{-1}$). Pollen influx is low (45,676–109,410 grains cm$^{-2}$yr$^{-1}$).

Zone 2 (107.5–58 cm, 18 samples, 1180–1550 C.E.). DTF taxa increase from 1180–1370 C.E. (75–95.3%) and persistently dominate the assemblage in abundances greater than 70% through to the end of this zone. High abundance DTF taxa include *Bursera simaruba* (6.5–45.6%), *Brosimum* (0–38.7%), *Terminalia* (4.1–16.4%), *Protium* (0–13%) and *Celtis* (0–16.7%). *Celtis* peaks at 1230 C.E. (16.7%). Coussapoa is persistent in low abundances between 1230–1530C.E (2.4–4.4%), at which point Anacardiaceae (3.3%) and *Mimosa* (3.3%) supersede. Herbaceous taxa decline and remain in low abundance until after 1440 C.E. (20.8–3.6%). From 1480 C.E., herbaceous taxa begin to increase through to the end of this zone (5.3–24.4%). *Capparis* (Caper) appears for the first time at c. 1500 C.E. accompanied by an increase in Poaceae. (2.6–6.7%). The abundance of non-DTF arboreal taxa remain in low abundance with the exception of a short-lived peak from *Roupala montana* at 1320 C.E. (23.7%). *Sporormiella* influx decreases from the beginning of this Zone (2.7 spores cm$^{-2}$yr$^{-1}$) and remains in low abundance between 1230–1480 C.E (0–0.7 spores cm$^{-2}$yr$^{-1}$). After 1480 C.E. *Sporormiella* influx increases substantially (0.7–2.7 spores cm$^{-2}$yr$^{-1}$). Macroscopic charcoal influx remains high until 1300 C.E. (30.6 particles cm$^{-2}$yr$^{-1}$) where it transitions down to a lower influx by 1400 C.E. (5.1 particles cm$^{-2}$yr$^{-1}$). Between 1400–1530 C.E. macroscopic charcoal influx remains comparatively low (0.1–7.9 particles cm$^{-2}$yr$^{-1}$). Pollen influx increases particularly after 1300 C.E. (266,775 grains cm$^{-2}$yr$^{-1}$) and is particularly high between 1440–1480 C.E. (1,413,110–1,333,889 grains cm$^{-2}$yr$^{-1}$).

Zone 3 (55.5–3.5 cm, 15 samples, 1550–2004 C.E.). DTF taxa decrease from 1560–1640 C.E. (83.6–48.4%) and increase from 1640–1750 C.E. (48.4–80%). The DTF assemblage is predominantly comprised of *Bursera simaruba* (6.4–36.7%), *Brosimum* (4.4–33.3%), *Protium* (0–9.6%), Leguminosae (2.2–7.1%), Anacardiaceae (1.4–10.1%), and *Rhus* (0–5.9%). Poaceae and Cyperaceae increase in abundance permanently after 1530 C.E. and 1600 C.E. Cyperaceae peaks at 1600 C.E. (15.8%) and 1640 C.E. (25.8%). DTF taxa decrease in abundance between 1750–1850C.E. reflected in the reduction of *Bursera simaruba* (36.7–19.5%), *Brosimum* (10–2.4%), and *Mimosa* (13.3–0%). Herbaceous taxa increase from 1750–1850 C.E. (16.7–46.3%). DTF abundance increases again from after 1850 C.E. (48.8%) through to 1970 C.E. (79.1%) with the increased abundance of *Bursera simaruba* (19.5–29.8%), *Protium* (4.9–10.5%), Leguminosae (2.4–11.9%), *Rhus* (2.4–10.5%), and *Terminalia* (0–6%). *Sporormiella* influx remains high between 1550–1570 C.E. (2–2.8 spores cm$^{-2}$yr$^{-1}$) and is high in 1650 C.E. (3.4 spores cm$^{-2}$yr$^{-1}$). After 1650 C.E. *Sporormiella* influx declines (3.4–0.8 spores cm$^{-2}$yr$^{-1}$). The abundance of macroscopic charcoal decreases towards the present from 1560–1970 C.E (3–0.03 particles cm$^{-2}$yr$^{-1}$). Pollen influx reduces overall (682,686–145,700 grains cm$^{-2}$yr$^{-1}$) but peaks at 1650 C.E. and (1,773,492 grains cm$^{-2}$yr$^{-1}$) and 1850 C.E. (1,391,104 grains cm$^{-2}$yr$^{-1}$).

PCA of the pollen assemblage displays taxa most associated with each Zone and explains 16.4% of variance on Axis 1 and 12.6% of variance on Axis 2 (Figure 7a). Zone 1 is most associated with *Protium*, Leguminosae, Compositeae and *Spananthe paniculate*; Zone 2 with *Bursera simaruba*, *Brosimum*, *Terminalia, Coussapoa* and *Celtis*; and Zone 3 with *Rhus, Trema, Cordia, Capparis, Curtella americana* and *Mimosa*.

CCA of the palynological assemblage delineates that macroscopic charcoal, *Sporormiella* and δ$^{18}$O are all statistically significant drivers of variation within the palynological assemblage. Macroscopic charcoal is most associated with Zones 1 and 2 along the first axis, orthogonally placed to δ$^{18}$O which is most associated with Zone 3 (Figure 7b). *Sporormiella* is associated in part with all Zones along the second axis.

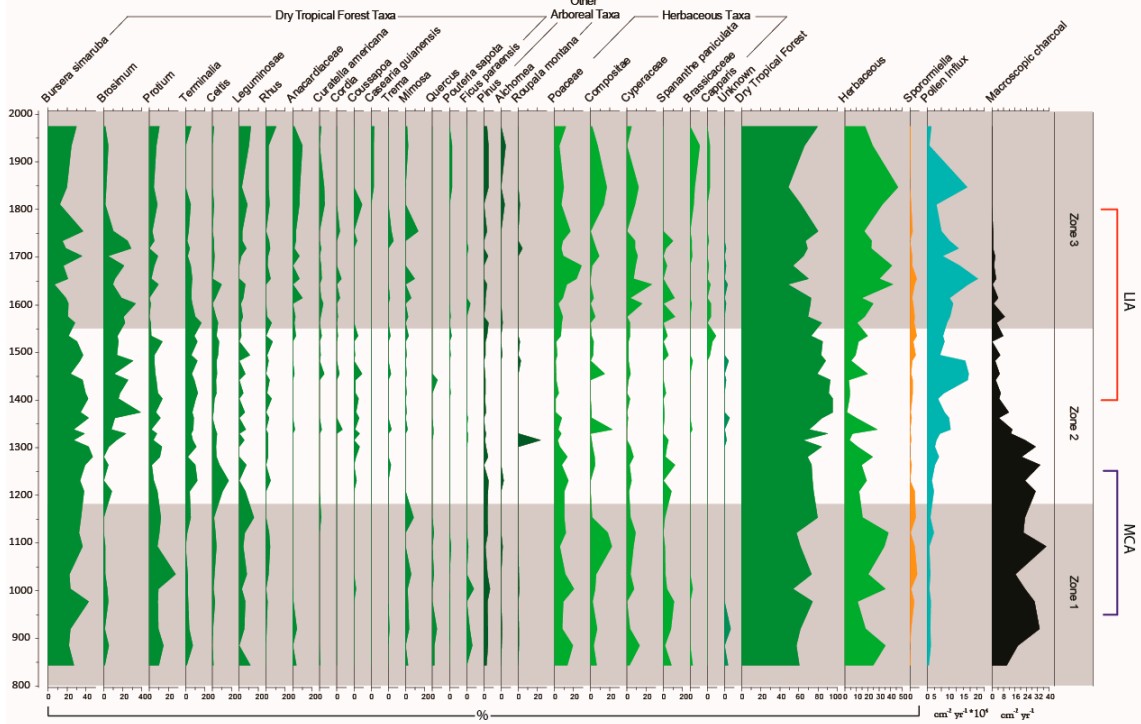

**Figure 6.** Palynological assemblage, *Sporormiella*, and macroscopic charcoal reconstructed from El Gancho, Nicaragua.

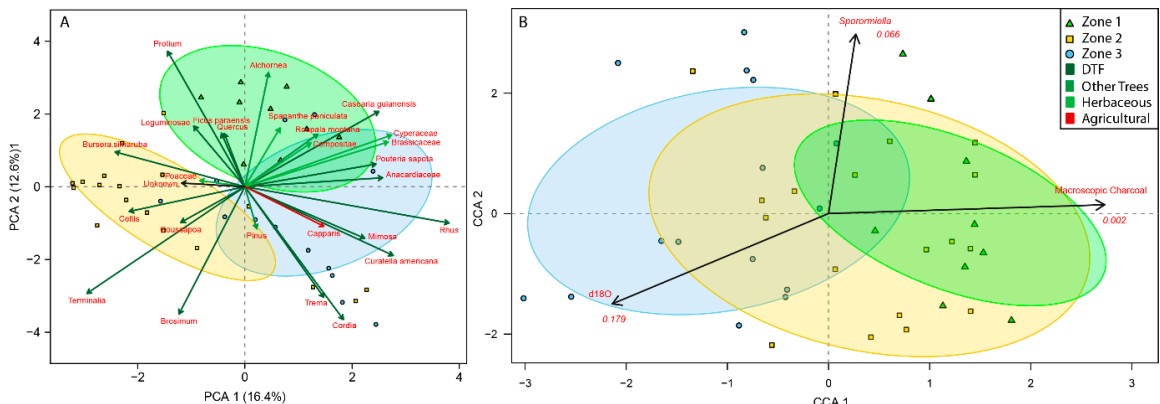

**Figure 7.** Principal component analysis of the palynological assemblage (**A**) Canonical correspondence analysis of the palynological assemblage (**B**) Zones derived from the broken stick model represented by ellipses at a confidence of 95%. Zone 1 = green triangles; Zone 2 = yellow squares; Zone 3 = blue circles.

## 4. Discussion

### 4.1. Hydroclimate Changes in the Nicaraguan Region of the Central American Dry Corridor during the Medieval Climate Anomaly and Little Ice Age

Interpretation of the diatom assemblage indicates that the Nicaraguan region of the CADC was wettest during the MCA (950–1250 C.E.) and drier during the transition to the LIA (1250–1400 C.E.) and throughout the LIA (1400–1800 C.E.). The transition between the MCA and LIA surrounding El Gancho is marked by a shift from wetter to drier conditions around 1300 C.E.

The high abundance of benthic and epiphytic diatom taxa between 840–1180 C.E. (Zone 1) indicates that wetter conditions were apparent in the region in the earliest part of this record during the MCA. The shallow basin that El Gancho currently occupies is poorly constrained and is surrounded by low lying topography (Figure 2). Subject to wetter conditions, El Gancho's basin is suggested to have expanded laterally, increasing in area but not necessarily depth. El Gancho is assumed to have been at its maximum expanse and depth during this period time. The high abundance of benthic and epiphytic diatom taxa, as well as Cyperaceae from the palynological assemblage, therefore, probably reflect the permanent inundation of water for a larger area by occupying the available habitat niche (e.g., Cyperaceae and *Potamogeton*) [27–29].

The time surrounding the transition between the MCA and LIA (c. 1200–1400 C.E.) leading into the first half of the LIA (1400–1600 C.E.) is defined by the replacement of benthic and epiphytic diatoms by the increasing abundance of diatom taxa indicative of changing conditions (e.g., *Pseudostaurosira brevistriata* and *Aulacoseira ambigua*), inferred to indicate a trend towards drier conditions (Figures 4 and 5). While the entire lake habitat is suitable for benthic and epiphytic diatoms to thrive, we suggest that these taxa were outcompeted by *Pseudostaurosira brevistriata* and *Aulacoseira* spp.

The rapid expansion and retraction of this water body between the wet season and the dry season is thought to have facilitated continuous biannual disturbance promoting a habitat more conducive for these two taxa to dominate the assemblage. This evidence further suggests that El Gancho retreated in area abandoning the large shallow expanse surrounding the central depocenter. After c. 1200 C.E. diatoms *Aulacoseira* spp. and *Pseudostaurosira brevistriata* comprise between 50–90% of the diatom assemblage. The succession of *Aulacoseira ambigua* at 1340–1600 C.E. is suggested to represent a rapid reduction in lake area, whereby the water occupying El Gancho retreated to the deepest part of its basin, abandoning the expansive shallow periphery. The succession back to facultatively planktonic taxa (*Pseudostaurosira brevistriata*) after 1600 C.E. is inferred to indicate that El Gancho continued to reduce in size, losing depth and available habitat for the planktonic diatom taxa to thrive (Figure 4). *Pseudostaurosira brevistriata* is a facultatively planktonic taxa and therefore we suggest that the available habitat niche provided conditions amiable to this taxa's success over the other taxa present, such as

*Aulacoseira ambigua*, in effect diluting the abundance of the other taxa. We previously explored the salinity and nutrient groupings for all diatom taxa and did not find conclusive trends associated with the most likely scenario presented in our manuscript.

The interpretation of the diatom record supports the $\delta^{18}O$ precipitation reconstruction presented by Stansell et al. [16] which also indicates wetter conditions for the MCA, particularly between c. 950–1300 C.E. and drier conditions thereafter. The response of the diatom assemblage to drier conditions during the transition between the MCA and LIA appears to lag behind the $\delta^{18}O$ presented in Stansell et al. [16] by c. 50 years; this could represent a biological lag response to hydroclimatic changes (Figure 4).

*4.2. How Did the Dry Tropical Forests Respond to These Wet–Dry Climatic Shifts?*

Three successional stages can be discerned from the last 1200 years of vegetation change in the DTF occupying the Asese peninsula (Figure 6). The first zone corresponds with wetter conditions from c. 800–1200 C.E., while the second (c. 1200–1550 C.E.) and third Zones (c. 1550–2004 C.E.) with relatively drier conditions (Figure 6).

The wettest period of the last 1200 years (c. 800–1250 C.E.) is primarily comprised of DTF taxa. The dominance of these taxa in the pollen diagram are taken to indicate a relatively closed canopy. Perhaps unusually with this forest type, there is evidence for intervals of greater burning. We suggest that this is in response to the wetter conditions from c. 800–1250 C.E. facilitating greater accumulation of vegetative biomass. This would have enabled larger fuel loads to develop and burn between seasons [96,100]. Fire in the CADC generally occurs in the transitions between the dry season and wet season as biomass dries out [112]. There is also an increased abundance of Poaceae, Compositeae, Cyperaceae and *Spananthe paniculate* during the MCA, which can represent burning facilitated by some openings in the canopy [112–114].

At the transition into relatively drier conditions (1250–1400 C.E.) there is a change in the composition of DTF taxa including the succession of *Brosimum, Terminalia and Celtis* (Figure 6). The highest abundance of these DTF taxa occurs under these drier conditions. In conjunction with this change in composition, reduced burning during this time occurred, probably associated with lower fuel loads. This reduction in burning (c. 1300 C.E.) appears to have resulted in an increasingly closed canopy, blocking light to the understory and the reduction in the presence of open ground taxa. Evidence of this closed canopy is also presented in the increase in the abundance of diatom flora *Aulacoseira spp.* which is suggested to be responsive to low-light levels, thriving in a well-developed forest with closed canopies [115].

*4.3. Anthropogenic Activity in the Dry Tropical Forests: Population and Land Management*

Pre-Columbian human occupation of the Asese peninsula has been documented from at least c. 600 C.E. through to 1250 C.E. with a hiatus during the Late Sapoa period c. 1100–1150 C.E. [116]. The nearby occupation of two municipalities within the Masaya region, as well as the occupation of Ometepe, was continuous from c. 200 B.C.E. through to the present day [52,53,117].

Despite this continuous Pre-Columbian occupation, there is no evidence of substantial deforestation in the pollen record of the Asese peninsula during this interval in time (c. 850–1522 C.E.); however, evidence of forest clearing is available from other nearby paleo-records. Deforestation on Ometepe Island, for example, from 775 C.E indicates that areas of Nicaragua were undergoing deforestation during Pre-Columbian times [118].

While there is no palynological evidence of Pre-Columbian agriculture (e.g., *Zea mays*), the abundance of fruiting trees and edible herbs are likely to have been utilised by local people for materials and sustenance. *Amaranthus spinosus*, Anacardiaceae, *Bursera simaruba, Brosimum, Capparis, Celtis*, Leguminosae and *Protium* are all examples of edible flora found in El Gancho, and were also eaten as part of the Pre-Columbian diet [119]. *Amaranthus spinosus* is a pioneering herbaceous plant with a similar nutritional value to that of spinach, and dominates the palynological record throughout [120].

The introduction of *Capparis* in the palynological record from c. 1500 C.E. possibly provides evidence for European contact. *Capparis* (commonly known as a caper) places its origins surrounding the Mediterranean in Europe, this taxon is not native to the Americas and would only have arrived with the Spanish conquistadors [121]. In addition, the reduction of select arboreal taxa, e.g., *Bursera simaruba*, and *Protium*, could reflect deforestation for building materials and the creation of pastures; however, this is also within the range of natural variability [122,123].

Evidence of herbivorous mammalian presence from c. 950–1250 C.E. aligns with the inferred wetter conditions brought about during the MCA. There is no current evidence for Pre-Columbian domestication of the native fauna, and therefore we suggest that greater vegetative growth during this period of time provided an increased food source for deer, peccary, and tapir allowing for their numbers to swell. Reductions in dung fungal spore abundance after the shift towards drier conditions, between the MCA and LIA, suggests that herbivorous mammals were largely absent from the Asese peninsula after 1250 C.E. until the arrival of the Spanish. Dung fungal spore evidence indicates that animal abundance on the Asese peninsula increased after the arrival of the Spanish through until 1650 C.E. This possibly represents the import of cattle brought into Granada (established 1524 C.E.) from Europe. These cattle would possibly have been kept on the Asese peninsula, a natural pen, until they could be sold at market [124]. The rise in population and clearance of land for agrarian practices leading up to 1800 C.E. may well therefore account for the apparent reduction of DTF taxa in and around the Asese peninsula from c. 1700–1850 C.E. During this time the abundance of DTF taxa reduce and economically important taxa such as Anacardiaceae increase.

### 4.4. Resilience of the Dry Tropical Forests Surrounding the Asese Peninsula

The reconstruction of the DTF surrounding the Asese peninsula, over the past 1200 years, show little evidence of vulnerability to climatic and anthropogenic impacts. While the composition of taxa comprising the DTF assemblage changes in response to the identified hydroclimatic and anthropogenic perturbations, the ecological structure of the DTF has remained remarkably resilient. High seasonal variation within the CADC has exposed the residing DTF to biannual disturbance since the modern configuration of the North American Monsoon c. 4000 B.C.E. [21]. Cole et al. [125] identifies high frequency disturbance as the most significant factor in determining resilience in tropical forests, which is clear from the reconstructions of DTF around the Asese peninsula. Biannual disturbance caused by seasonal variation over the last four millennia is the likely reason why these DTF are so resilient, allowing these DTF to adapt and increase their resilience over time [126].

Studies of satellite data [127] and meta-analyses [35] examining resilience of vegetation in this region in recent decades to climatic perturbations suggest that DTF are sensitive to changes in precipitation/drought intensity, frequency, and/or timing; however, the results presented from El Gancho suggest that the DTF around the Asese peninsula are not particularly sensitive to climatic and anthropogenic fluctuations, at least not over the past 1200 years. While there is some evidence for the reorganisation of taxa between the palynological Zones, likely responding to more frequent or severe drought or variability in rainfall [128–130], the palynological abundance of DTF taxa in the record from El Gancho never fall below 50%, and exceeds 90% from c. 1350–1420 C.E., when fire reduces and climate switches to overall drier conditions (Figure 6). Further, these findings suggest that the vegetation surrounding the Asese peninsula has not been subject to a tipping point leading to a switch to a different stable state and vegetation type during the time frame studied [30].

Specific DTF taxa such as *Bursera simaruba* and *Brosimum*, which are prominent in the palynological record from El Gancho, are known to have a high wood density comprised of greater cell wall material and narrow vessels. These limit their hydraulic efficiency but increase their resistance to drought-induced water scarcity [131,132]. High wood density has previously been demonstrated to be an important factor in enabling persistence of woody taxa during intervals of drought [130]. These species may also resist extended periods of drought by tapping into subsoil water reserves with their deep roots [35,133,134].

## 5. Conclusions

The results from this research suggest that DTF have been highly resilient to past climatic and anthropogenic perturbations over the past 1200 years (Table 2). Interpretations of changes in climate between the MCA and LIA support wetter conditions during the MCA (950–1250 C.E.) and drier conditions in the transition between the MCA and LIA (1250–1400 C.E.) and the LIA (1400–1850 C.E.). The proxy analysis of the diatom record independently supports this interpretation of the $\delta^{18}O$ first presented by Stansell et al. [16].

Changes in DTF composition appear to be linked to the abundance and intensity of fire with a high amount of burning during the MCA and low burning thereafter c. 1300 C.E. Pre-Columbian anthropogenic impacts on DTF are not detected in the palynological record; however, DTF taxa are noted to decline slightly after European contact (1522 C.E.).

After European contact, and through until c. 1650 C.E., there is an increase in dung fungal spore abundance which possibly represents the increased import of cattle brought into the city of Granada.

It is apparent that over the past 1200 years there have been significant fluctuations in climatic and anthropogenic impacts; however, evidence from El Gancho suggests that changes in the structure of DTF surrounding the Asese peninsula have responded with little consequence. Drought resilient traits of the dominant DTF taxa (e.g., *Bursera simaruba* and *Brosimum*) including high wood density and long roots may have helped this community resist climatic and anthropogenic impacts over the past 1200 years.

Rapid agricultural growth from c. 1800 C.E. has allowed for larger populations to be supported; however, it has also increased the reliance on agriculture. Areas that were once covered by DTF that have since been converted to pastures and plantations do appear to be highly threatened by seasonal hydrological fluctuations. An agrarian based society in the CADC is therefore suggested to be far more vulnerable to hydroclimatic changes than a society that primarily relies upon hunting and gathering for sustenance.

**Table 2.** Chronology of palynological zones, archaeological periods [52,53,117] local populations [45–48,51,116,117] and climatic shifts [16].

| Age (C.E.) | Zone | Archaeological Period | | | Population | | | Hydroclimate | | | DTF | Fire |
|---|---|---|---|---|---|---|---|---|---|---|---|---|
| | | Ometepe | Rivas | Masaya | Nicaragua | Masaya | El Rayo | Diatom | $\delta^{18}O$ | Regional | | |
| 2000 | 3 | Post-Conquest | Post-Conquest | Post-Conquest | 6,000,000 | | | Drier | Drier | LIA | | Low |
| 1900 | | | | | 1,000,000 | | | | | | | |
| 1800 | | | | | 81,000 | | | | | | | |
| 1700 | | | | | 61,000 | | | | | | | |
| 1600 | | | | | | | | | | | | |
| 1500 | 2 | Santa Ana | Alta Gracia | Ometepe | | 3630 | | | | Transition | | |
| 1400 | | | | | | | | | | | | |
| 1300 | | San Lazaro | Las Lajas | | 100,000–1,000,000 | | | Wetter | Wetter | MCA | | High |
| 1200 | 1 | La Paloma | La Virgen | Sapoa | | 4070 | Occupied | | | | | |
| 1100 | | | | | | | | | | | | |
| 1000 | | | | | | | | | | | | |
| 900 | | Gato | Apompua | | | | | | | | | |
| 800 | | | | | | | | | | | | |

**Supplementary Materials:** The following are available online at http://www.mdpi.com/2571-550X/2/3/25/s1, Table S1: Images of diatom taxa at 400x magnification identified from El Gancho, Table S2: Images of palynological taxa at 400x magnification identified from El Gancho, Figure S1: Amaranthus spinosus field specimen. Collected on 20 May 2015 from 11°54′22.4″N 85°55′08.4″W.

**Author Contributions:** W.J.H. conceptualized the presented idea with input from N.S., S.N., and K.J.W. W.J.H. conducted all palaeoecological lab. work, statistical analysis, and age-depth modelling. S.N. and K.J.W. verified

the analytical methods. N.S., S.N., and K.J.W. helped supervise the project. W.J.H. drafted the final manuscript. All authors discussed the results contributing to the final manuscript.

**Funding:** This research was funded by the Natural Environment Research Council of the United Kingdom NE/L002612/1, US National Science Foundation EAR-1502989, the Geological Society of America, and St Edmund Hall, University of Oxford.

**Acknowledgments:** I would like to give thanks to Julio B. Gomez Martinez, Veronica Lacayo de Gomez and Julio B. Gomez Lacayo, and Maurice Hodgson for their support while conducting field work in Nicaragua, and Gillian Petrokofsky, Peter Long and Carol Adolf from the Oxford Long-Term Ecology Lab., and Ian Matthews from Royal Holloway.

**Conflicts of Interest:** The authors declare no conflicts of interest.

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
