# Peer review of "The Apparent Resilience of the Dry Tropical Forests of the Nicaraguan Region of the Central American Dry Corridor to Variations in Climate Over the Last C. 1200 Years"

_quaternary, doi:10.3390/quat2030025_

Round 1

Reviewer 1 Report

In my opinion the authors present an interesting story about the resilience of the dry forest and about changes in the aquatic ecosystem during this period of global climatic changes. The article is of significance since “paleo” studies in this area are scarce. I recommend that the authors explore in more depth their pollen, charcoal and diatom records, before interpreting them only as a response to climate variability; the stratigraphic variations in diatoms, pollen and charcoal could have an alternative explanation. An important aspect to consider is that the pollen signal could be biased by human intervention, and thus, the forest is an already managed ecosystem and thus difficult to assess its resilience as a non-impacted forest (resilience is then measured under human intervention). Can the charcoal be reflecting anthropogenic clearance? A good discussion on human impact is necessary (I suggest to start the Discussion with section 4.3.). Also, could it be that the changes in temperature and precipitation were not that drastic during MCA and LIA? Could the authors estimate how wet or how dry conditions were during these two events based on the 18O record?  

It would be informative to add a discussion on the litho-stratigraphy of the core, possible hiatuses, the coring site, and the water balance of the swamp today or the effect of groundwater.

            Perhaps, as a suggestion, section 4.3. can be discussed upfront, followed by sections 4.1 and 4.2. that could be merged and discussed under ecological changes in the aquatic and terrestrial ecosystems, supported by PC/CCA analyses, and then compared it against the 18O signal.

Here detail comments on the discussion:

Section 4.1.

The authors infer hydroclimatic changes wet/dry periods on the diatom diagram more than the 18O signal. Having this record, I suggest that the authors explore further deep other reasons for the observed variations between planktic, benthic and epiphytic taxa. The authors interpret the abundance of benthic and epiphytic taxa as indicative of extension of the littoral and thus wetter conditions. While this is true, it also depends on the coring site. A core from the pelagic zone or deepest part of the swamp, will have planktic and littoral taxa (epiphytic and benthic),while a core from the littoral, most likely will record dominance of epiphytic and benthic. It would be important to disclose the coring site, in order to interpret better the rations plankton/benthos.

In Zone 1 there are species tolerant to alkaline and or saline conditions, e.g. Mastogloia, Anomoeoneis sphaerophora, Brachysira vitrea, and others favoured under high nutrient waters (Cyclotella meneghiniana, Nitzschia amphibia), or tolerant to a wide salinity range like Nitzschia frustulum (see Reimer 1964 https://digitalcommons.butler.edu/cgi/viewcontent.cgi?referer=https://www.google.com/&httpsredir=1&article=1231&context=botanical). To me, before arriving to a wet/dry climatic interpretation, I suggest to explore further the ecological requirements of the dominant diatoms as there are indications that water quality (alkalinity, conductivity and or nutrients) played an important role. It seems to me that Zone 1 could be also interpreted -based on the dominance of saline tolerant taxa, and higher nutrient states in the water-, as indicative of drier conditions. It would be very good to compare the lithology vs the diatom record (checking the detrital vs organic ratio of the sediments, grain size and presence of evaporitic minerals, etc.).

The interpretation of Zone 2 needs some revision, in particular the interpretation of Aulacoseira ambigua as indicative of disturbance conditions, or a reference is needed, and being favoured under low light levels. This can be true, but before that it throives under higher nutrient concentrations. This zone contains, although, in low proportion, the highest diversity of Aulacoseira spp. and the almost continuous presence of Cyclotella meneghiniana. An alternative explanation to lake encroachment is, higher water levels and more permanent water conditions, as these Aulacoseira are rather robust diatoms that require some turbulence, and prefer mesotrophic waters (see Frizt et al. 2019). Since this would suggest wetter conditions, and the 18O signal points to drier conditions, would it be possible that there is a groundwater effect? Again, it would be good to know where was the core extracted, the bathymetry of the swamp, water balance, and if there is a depocentre as suspected by the authors.

In summary, I suggest that the authors explore a bit more other ecological and limnological variables (nutrients, alkalinity and ground water) to explain the stratigraphic variations in the diatom record (and to add the references for the ecology of the diatoms). Inferring past hydroclimatic conditions based only on habitat could lead to misinterpretations.   

Section 4.2.

The authors again interpret the stratigraphic variations of the pollen record and charcoal in terms of dry/wet conditions, and since there is little variation in the record, the authors conclude that the forest has been resilient to climatic variations. However, since human occupation in the region dates back to some time prior to the record presented here, it is highly possible that this pollen signal is being affected by human impact, and making it difficult to assess the “natural” response of the DF to changes in precipitation. In fact, the abundance of macrocharcoal in Zone 1 and part of Zone 2, could be anthropogenic. Maybe I missed it but the authors did not consider human impact behind the fire/pollen signal. For instance, could the increase in Brosimum, be due to human alteration?

            Also the relationship between burning and canopy needs to be better discussed, as it is not completely clear. Can be open ground taxa be absent due to a preferable aliment selection by smaller mammals?

Section 4.3.

Is the increase in fungal spores after Spanish arrival significant? At least its increase after 1650CE is not clear from the figure, a similar modest increase is present Zone 1; how could this be explained?

I suggest to start with this section in the discussion.

Other minor comments:

In most cases B.C.E and C.E. or units (e.g. mm, km) follow the number without a space

Line 51 “and. These” must be corrected

Line 103 correct the degree symbol for temperature

Line 104 greater potential “of’?

Line 108 correct citation number

Table 1. correct 14C to reflect chemical nomenclature

Section 2.3. Lines 198-199 not clear what the authors mean by “The diatom samples were matched to the sampling resolution of the palynological data through linear interpolation for ease of comparative analysis.”

Add few lines on the magnification and microscope usage for diatom analyses (to be consistent with the reporting of the pollen analyses)

Line 235 correct “interpellation”

Line 244 correct “sub samples”

Section 3.2. I suggest to revise the title as it does not really reflect paleoecological trends. As it is, is only the description of the abundances.

Line 291 add “s” to Figure

For Figures 4 and 6: were all the taxa counted presented in this figure? Or is it a selection? E.g. those that have counting >5?

There is overall very little discussion and integration of on the results from CCA and PCA on the Discussion section

Line 389: I suggest to revise the sentence “Perhaps unusually with this forest type, there is evidence for intervals of greater burning.”

Line 397 “and” is in italics

The diatom diagram

1. Update of names and authors or at least to be consistent (Pseudostaurosira vs Fragilaria; Navicula pupula (Sellaphora pupula).

2. Were all the countings presented or did the authors selected those above 5% or 2%

I suggest to revise the epiphytic character of Nitzschia amphibia

Supplementary material:

There is no scale on the diatoms, however, I suggest to check the morphology of:

Aulacoseira. ambigua, I am not sure it is A. ambigua.

Navicula longicephala, check Encyonopsis cesatii

Navicula pennsylvanica, check Diadesmis confervacea

Author Response

The apparent resilience of the dry tropical forests of the Nicaraguan region of the Central American dry corridor to variations in climate over the last c.1200 years

The authors would like to thank all three reviewers for their insightful and constructive comments for improving this manuscript and hope that they find the below responses and additions to the manuscript satisfactory.

Reviewer 1

In my opinion the authors present an interesting story about the resilience of the dry forest and about changes in the aquatic ecosystem during this period of global climatic changes. The article is of significance since “paleo” studies in this area are scarce. I recommend that the authors explore in more depth their pollen, charcoal and diatom records, before interpreting them only as a response to climate variability; the stratigraphic variations in diatoms, pollen and charcoal could have an alternative explanation. An important aspect to consider is that the pollen signal could be biased by human intervention, and thus, the forest is an already managed ecosystem and thus difficult to assess its resilience as a non-impacted forest (resilience is then measured under human intervention).

1.      Can the charcoal be reflecting anthropogenic clearance? A good discussion on human impact is necessary (I suggest to start the Discussion with section 4.3.).

a.      While the charcoal record could partially reflect human impact the coincidence of high burning and lower burning with climatic variation is so profound that climate is the most likely scenario. Due to a scant archaeological record we cannot speculate further on human interactions.

2.      Also, could it be that the changes in temperature and precipitation were not that drastic during MCA and LIA?

a.      Changes in temperature and precipitation during the MCA and LIA are highly debated in the literature, we cannot suggest absolutes from our data and instead focus on trends.

3.      Could the authors estimate how wet or how dry conditions were during these two events based on the 18O record?

a.      It is not possible to estimate absolute precipitation from the data we have collected.

4.      It would be informative to add a discussion on the litho-stratigraphy of the core, possible hiatuses, the coring site, and the water balance of the swamp today or the effect of groundwater.

a.      A description of the stratigraphy of the core and LOI is published in the supplementary information of the first El Gancho paper (Stansell et al. 2013). We have also added in a short sentence to clarify this point “The upper ~168 cm of the core contains massive, very dark brown to black, organic-rich sediments with no notable changes in the stratigraphy.

5.      Perhaps, as a suggestion, section 4.3. can be discussed upfront, followed by sections 4.1 and 4.2. that could be merged and discussed under ecological changes in the aquatic and terrestrial ecosystems, supported by PC/CCA analyses, and then compared it against the 18O signal.

a.      We would like to thank the reviewer for their suggestion; however, the focus of this paper is on the resilience of the DTF and not the archaeological record. While the anthropological aspect inferred from of our record may be of interest to the archaeological community all authors feel that the focus should remain on the ecology with drivers of and responses of the DTF vegetation.

6.      The authors infer hydroclimatic changes wet/dry periods on the diatom diagram more than the 18O signal. Having this record, I suggest that the authors explore further deep other reasons for the observed variations between planktic, benthic and epiphytic taxa. The authors interpret the abundance of benthic and epiphytic taxa as indicative of extension of the littoral and thus wetter conditions. While this is true, it also depends on the coring site. A core from the pelagic zone or deepest part of the swamp, will have planktic and littoral taxa (epiphytic and benthic),while a core from the littoral, most likely will record dominance of epiphytic and benthic. It would be important to disclose the coring site, in order to interpret better the rations plankton/benthos.

a.      Please see lines 75 – 76 and the response to comment 3 for Reviewer 3.

7.      In Zone 1 there are species tolerant to alkaline and or saline conditions, e.g. Mastogloia, Anomoeoneis sphaerophora, Brachysira vitrea, and others favoured under high nutrient waters (Cyclotella meneghiniana, Nitzschia amphibia), or tolerant to a wide salinity range like Nitzschia frustulum (see Reimer 1964 https://digitalcommons.butler.edu/cgi/viewcontent.cgi?referer=https://www.google.com/&httpsredir=1&article=1231&context=botanical). To me, before arriving to a wet/dry climatic interpretation, I suggest to explore further the ecological requirements of the dominant diatoms as there are indications that water quality (alkalinity, conductivity and or nutrients) played an important role. It seems to me that Zone 1 could be also interpreted -based on the dominance of saline tolerant taxa, and higher nutrient states in the water-, as indicative of drier conditions. It would be very good to compare the lithology vs the diatom record (checking the detrital vs organic ratio of the sediments, grain size and presence of evaporitic minerals, etc.).

a.       We previously explored the salinity and nutrient groupings for all diatom taxa and did not find conclusive trends associated with the most likely scenario presented in our manuscript.

b.      The upper ~168 cm of the core contains massive, very dark brown to black, organic-rich sediments with no notable changes in the stratigraphy. See addition to manuscript.

c.      There is also limited variation in the magnetic susceptibility – please see the supplementary information for Stansell et al. 2013

8.      The interpretation of Zone 2 needs some revision, in particular the interpretation of Aulacoseira ambigua as indicative of disturbance conditions, or a reference is needed, and being favoured under low light levels. This can be true, but before that it throives under higher nutrient concentrations. This zone contains, although, in low proportion, the highest diversity of Aulacoseira spp. and the almost continuous presence of Cyclotella meneghiniana. An alternative explanation to lake encroachment is, higher water levels and more permanent water conditions, as these Aulacoseira are rather robust diatoms that require some turbulence, and prefer mesotrophic waters (see Frizt et al. 2019). Since this would suggest wetter conditions, and the 18O signal points to drier conditions, would it be possible that there is a groundwater effect? Again, it would be good to know where was the core extracted, the bathymetry of the swamp, water balance, and if there is a depocentre as suspected by the authors. In summary, I suggest that the authors explore a bit more other ecological and limnological variables (nutrients, alkalinity and ground water) to explain the stratigraphic variations in the diatom record (and to add the references for the ecology of the diatoms). Inferring past hydroclimatic conditions based only on habitat could lead to misinterpretations.   

a.      Please see the revised text for Aulacoseira ambigua as indicative of changing conditions.

b.      The lake sits upon a raised peninsula around 10m above Cocibolca. To our knowledge, there is no groundwater effect upon the closed basin – see Stansell et al. 2013.

9.      The authors again interpret the stratigraphic variations of the pollen record and charcoal in terms of dry/wet conditions, and since there is little variation in the record, the authors conclude that the forest has been resilient to climatic variations. However, since human occupation in the region dates back to some time prior to the record presented here, it is highly possible that this pollen signal is being affected by human impact, and making it difficult to assess the “natural” response of the DF to changes in precipitation. In fact, the abundance of macrocharcoal in Zone 1 and part of Zone 2, could be anthropogenic. Maybe I missed it but the authors did not consider human impact behind the fire/pollen signal. For instance, could the increase in Brosimum, be due to human alteration?

a.      Current available evidence (presented in our record) suggests that impacts to the forests for the last 1200 years are primarily natural. It is possible that the forests have already been impacted by humans prior to 800C.E.; however, we lack a longer record to draw upon these uncertainties. The archaeological record is scant for this region and we await further research to explore potential human interactions and impacts to these forests.

b.      We suggest that Brosimum can be explained by the changes in climate and while we accept that humans may have altered the abundance of particular taxa there is currently no supporting archaeological evidence to suggest strong anthropogenic impact. There are no known pre-Columbian state level societies in Nicaragua.

10.   Also the relationship between burning and canopy needs to be better discussed, as it is not completely clear. Can be open ground taxa be absent due to a preferable aliment selection by smaller mammals?

a.      The greater accumulation of vegetative biomass can allow for larger fuel loads to develop and burn between seasons. This would facilitate more intensive burning which would impact the canopy.

b.      There is never an absence of open ground taxa in this record. Smaller animals are likely to be present but are not thought to impact the assemblage in any meaningful way.

11.   Is the increase in fungal spores after Spanish arrival significant? At least its increase after 1650CE is not clear from the figure, a similar modest increase is present Zone 1; how could this be explained?

a.      Please see the addition to the text – Lines 437 – 450.

12.   In most cases B.C.E and C.E. or units (e.g. mm, km) follow the number without a space

a.      Can the editor please advise on the convention for this journal.

13.   Line 51 “and. These” must be corrected

a.      Thank you  - Correction implemented.

14.   Line 103 correct the degree symbol for temperature

a.      Thank you  - Correction implemented.

15.   Line 104 greater potential “of’?

a.      Thank you  - Correction implemented.

16.   Line 108 correct citation number

a.      Thank you  - Correction implemented.

17.   Table 1. correct 14C to reflect chemical nomenclature

a.      Thank you  - Correction implemented.

18.   Section 2.3. Lines 198-199 not clear what the authors mean by “The diatom samples were matched to the sampling resolution of the palynological data through linear interpolation for ease of comparative analysis.”

a.      Point sampling was taken at intervals slightly apart from the palynological and charcoal analysis and so to allow for statistical comparison these were matched.

19.   Add few lines on the magnification and microscope usage for diatom analyses (to be consistent with the reporting of the pollen analyses)

a.      Thank you  - Correction implemented.

20.   Line 235 correct “interpellation”

a.      Thank you  - Correction implemented.

21.   Line 244 correct “sub samples”

a.      Thank you  - Correction implemented.

22.   Section 3.2. I suggest to revise the title as it does not really reflect paleoecological trends. As it is, is only the description of the abundances.

a.      The authors believe that the title reflects the proxy which is palaeoecological.

23.   Line 291 add “s” to Figure

a.      Thank you  - Correction implemented.

24.   For Figures 4 and 6: were all the taxa counted presented in this figure? Or is it a selection? E.g. those that have counting >5?

a.      All taxa identified and counted are presented – please see raw data in supplementary file.

25.   There is overall very little discussion and integration of on the results from CCA and PCA on the Discussion section.

a.      The PCA and CCA have been used to illustrate associations along gradients as discussed in the manuscript.

26.   Line 389: I suggest to revise the sentence “Perhaps unusually with this forest type, there is evidence for intervals of greater burning.”

a.      The Authors believe this sentence does not need revision.

27.   Line 397 “and” is in italics

a.      Thank you  - Correction implemented.

28.   Update of names and authors or at least to be consistent (Pseudostaurosira vs FragilariaNavicula pupula (Sellaphora pupula).

a.      The authors believe they have been consistent using the taxonomic identification keys referenced in the text.

29.   Were all the countings presented or did the authors selected those above 5% or 2%

a.      All taxa identified and counted are presented – please see raw data in supplementary file.

30.   I suggest to revise the epiphytic character of Nitzschia amphibia

a.      Please see grouping based upon: Miranda, M.C., 1997. The last glacial maximum in the basin of Mexico: the diatom record between 34,000 and 15,000 years BP from Lake Chalco. Quaternary International43, pp.125-136.

31.   Supplementary material: There is no scale on the diatoms, however, I suggest to check the morphology of:

a.      Aulacoseira. ambigua, I am not sure it is A. ambigua.

                                                  i.     The Authors have checked this taxa and believe this is Aulacoseira ambigua.

b.     Navicula longicephala, check Encyonopsis cesatii

                                                  i.     The Authors have checked this taxa and believe this is Navicula longicephala.

c.      Navicula pennsylvanica, check Diadesmis confervacea

                                                  i.     The Authors have checked this taxa and believe this is Navicula pennsylvanica

Reviewer 2 Report

The apparent resilience of the dry tropical forests of the Nicaraguan region of the Central American dry corridor to variations in climate over the last c.1200 years

The manuscript explores the resilience of dry tropical forest in Centro America by combining multiple lines of evidence, including diatom, pollen and Sporomiella. The results of the manuscript could provide with important information regarding the resiliency of this ecosystem to the current climate crisis. However certain aspects of the manuscript need to be developed in more detail.

Introduction

Paragraph 1. The justification of the study is a little weak. It is not clear why Dry Tropical Forests are important to study and how they are expected to respond to current climate warning. Moreover, the importance of reconstructing the paleoenvironment needs to be highlighted and how it would contribute to understanding dry forests.

How do the authors define ecosystem resilience? What did you expect to find in the paleorecord during extreme climatic events?

Line 73 – The authors indicate that 3 sites have been studied but they only provided only detailed information from only one site. Do the other two studies show similar results? Provide more detailed information.

As the authors mention, the region has been inhabited by human groups that modified the landscape extensively. I consider important to include a section that reviews the archeological evidence and the role of human groups on the modification of the vegetation.

Line 106 – Avoid using quotes

Given that the study is also focused on reconstructing presence of large mammals. It would be important to also describe the native fauna of the region.

Results

Describe the stratigraphy of the sediment core.

Discussion

The interpretation of the diatom and pollen analysis are briefly correlated. Does changes in the diatom composition also reflects resilience of the aquatic community?

It is possible that the large abundance of Amaranthus in the pollen record could suggest intense human disturbance in the region. Many species of Amaranthus are hyperabundant in human disturbed habitant and are able to easily colonized recently disturbed soils. Given that Amaranth is also an edible species, could it be possible that the large abundance of the species indicates management of the landscape by human groups?

Why does Capparis indicate European contact?

The abundance of Sporomiella is also relatively high by the end of the Zone 3. What does it mean?

Section 4.4 Resilience of DTF

Although the pollen reconstruction does not reflect significant changes in vegetation composition during extreme dry/wet periods and due to land-use intensification, it is important to mention the limitations of the study before suggesting that the high resilience of this ecosystem.

Suggesting that dry tropical forests are resilient to perturbation should be back up by additional evidence. Do other studies support similar results in Mesoamerica?

Could it be possible that mostly resilient and common taxa are represented in the pollen record?, and thus masking the effects extreme climatic events and disturbance?

Conclusion

Due to the importance of the study to predict the impact of current climate change and high rates of habitat transformation, consider including a section at the end of the conclusion regarding the implications of the study.

Figure 1. It would be very illustrative if the distribution of the DTF is included in both maps.

Figure 6. Include pollen concentration of Amaranthus

Author Response

The apparent resilience of the dry tropical forests of the Nicaraguan region of the Central American dry corridor to variations in climate over the last c.1200 years

The authors would like to thank all three reviewers for their insightful and constructive comments for improving this manuscript and hope that they find the below responses and additions to the manuscript satisfactory.

Reviewer 2

The manuscript explores the resilience of dry tropical forest in Centro America by combining multiple lines of evidence, including diatom, pollen and Sporomiella. The results of the manuscript could provide with important information regarding the resiliency of this ecosystem to the current climate crisis. However certain aspects of the manuscript need to be developed in more detail.

1.      Paragraph 1. The justification of the study is a little weak. It is not clear why Dry Tropical Forests are important to study and how they are expected to respond to current climate warning. Moreover, the importance of reconstructing the paleoenvironment needs to be highlighted and how it would contribute to understanding dry forests.

a.      We would like to thank Reviewer 2 for their concern and hope that we have clarified their point below. The justification for this study directly relates to the current impact of climate change to the biota (plants, animals, and people) of the Central American Dry Corridor region (highlighted in lines 31 – 39) and how periods of drought have resulted in significant losses in food production, which has threatened 2.2 million people with food and water insecurity. We have clarified in text (line 38) that the reason we focussed on the Dry Tropical Forests was because this is the dominant vegetation type for this region. Please see the new Figure (Figure 1) which highlights this assertion. The authors do not feel that further explanation is required to describe an area that is most entirely classified as Dry Tropical Forest.

2.      How do the authors define ecosystem resilience? What did you expect to find in the paleorecord during extreme climatic events?

a.      We are defining resilience by the capacity of a system (biotic or abiotic) to be perturbed while remaining in the same state. Please see addition to text (lines 90 – 91).

b.      We expected to find that either the: “DTF (i) will be sensitive to climatic change because they are already limited by water and are close to a habitat threshold; or (ii) they will be resilient to changes in rainfall because they are adapted to predictable seasonal drought. Similarly, DTF will either be (i) vulnerable to anthropogenic activities, such as agrarian practices, land conversion and population density because of land use change; or (ii) they will be resilient to anthropogenic factors due to composition adaptation over time…” (lines 85 – 90).

3.      Line 73 – The authors indicate that 3 sites have been studied but they only provided only detailed information from only one site. Do the other two studies show similar results? Provide more detailed information.

a.      We would like to thank Reviewer 2 for advising on the clarity of these three studies. All sites are from the Ahuachapan region of El Salvador and show similar results - the references in text have been updated to reflect these findings.

4.      As the authors mention, the region has been inhabited by human groups that modified the landscape extensively. I consider important to include a section that reviews the archeological evidence and the role of human groups on the modification of the vegetation.

a.      While we have utilised the limited archaeological records available (particularly for population estimates), the currently published archaeological records are scant. Until further peer reviewed archaeological records are produced from this region all authors do not feel it is appropriate to speculate on anthropological interactions with these forests.

5.      Line 106 – Avoid using quotes

a.      Quotes have been removed.

6.      Given that the study is also focused on reconstructing presence of large mammals. It would be important to also describe the native fauna of the region.

a.      We have added in a small section describing the dominant large mammals for this region during pre-Colombian times. Current animals found on the Asese peninsula are described in the last paragraph of section ‘1.4 El Gancho’ line 142.

7.      Describe the stratigraphy of the sediment core.

a.      A description of the stratigraphy of the core and LOI is published in the supplementary information of the first El Gancho paper (Stansell et al. 2013). We have also added in a short sentence to clarify this point “The upper ~168 cm of the core contains massive, very dark brown to black, organic-rich sediments with no notable changes in the stratigraphy.

8.      The interpretation of the diatom and pollen analysis are briefly correlated. Does changes in the diatom composition also reflects resilience of the aquatic community?

a.      The focus of this paper is to assess the resilience of the Dry Tropical Forests and not diatom community dynamics. Diatoms were used to identify hydroclimatic changes and lake level/shape fluctuations only.

9.      It is possible that the large abundance of Amaranthus in the pollen record could suggest intense human disturbance in the region. Many species of Amaranthus are hyperabundant in human disturbed habitant and are able to easily colonized recently disturbed soils. Given that Amaranth is also an edible species, could it be possible that the large abundance of the species indicates management of the landscape by human groups?

a.      While it is possible that a large abundance of Amaranthus in the pollen record could suggest intense human disturbance in the region, just as it is possible that some species could have been cultivated for human consumption, there is currently insufficient evidence to make assumptions using the palynological sequence alone. With future archaeological research perhaps some of these assertions could be realised.

10.   Why does Capparis indicate European contact?

a.      Capparis (commonly known as a caper) places its origins surrounding the Mediterranean in Europe (see reference 122). This taxon is not native to the Americas and would only have arrived with the Spanish conquistadors. Please see addition to the manuscript.

11.   The abundance of Sporomiella is also relatively high by the end of the Zone 3. What does it mean?

a.      Sporormiella has almost completely disappeared by the end of Zone 3. This is likely to suggest more limited herbivorous mammalian presence on the peninsula. If Reviewer 2 was referring to Zone 1 then please see the addition to the text.

12.   Although the pollen reconstruction does not reflect significant changes in vegetation composition during extreme dry/wet periods and due to land-use intensification, it is important to mention the limitations of the study before suggesting that the high resilience of this ecosystem.

a.      Limitations of this study primarily surround the sample resolution and uncertainty in the age-depth model. This is a common variable in palaeoecological reconstructions, well known, and accepted in this field.

13.   Suggesting that dry tropical forests are resilient to perturbation should be back up by additional evidence. Do other studies support similar results in Mesoamerica?

a.      There are currently no other publications that have been conducted in the Dry Tropical Forest across Mesoamerica - beyond what is already discussed in the introduction (references 27 – 29). This is one of the key findings that makes this papers publication so pertinent.

14.   Could it be possible that mostly resilient and common taxa are represented in the pollen record?, and thus masking the effects extreme climatic events and disturbance?

a.      Common taxa are by definition going to be most represented in the palynological record. Resilient taxa are more likely to persist through time as they survive a disturbance event, please refer to our initial hypotheses: “DTF (i) will be sensitive to climatic change because they are already limited by water and are close to a habitat threshold; or (ii) they will be resilient to changes in rainfall because they are adapted to predictable seasonal drought. Similarly, DTF will either be (i) vulnerable to anthropogenic activities, such as agrarian practices, land conversion and population density because of land use change; or (ii) they will be resilient to anthropogenic factors due to composition adaptation over time…” (lines 85 – 90).

b.      Persistence of resilient taxa could mask some of the effects of extreme climatic events; however, this statement just further supports our conclusions that the Dry Tropical Forests of Central Pacific Nicaragua have been highly resilient for at least the last 1200 years.

15.   Due to the importance of the study to predict the impact of current climate change and high rates of habitat transformation, consider including a section at the end of the conclusion regarding the implications of the study.

a.      The implications for the resilience of the Dry Tropical Forests to current climate change from this study implies that we do not need to be concerned with the impacts of climatic changes for this biome; however, with only limited studies from the Dry Tropical Forests we do not have sufficient spatial coverage to draw definitive implications. All authors do not feel it appropriate to making sweeping statements that could be picked up by policy makers without further research for this region.

16.   Figure 1. It would be very illustrative if the distribution of the DTF is included in both maps.

a.      Thank you for this suggestion, please see the new figure (Figure 1).

17.   Figure 6. Include pollen concentration of Amaranthus

a.      We thank Reviewer 2 for this suggestion; however, all authors feel that adding in the concentrations for Amaranthus would not contribute towards the overall narrative and would further clutter the diagram. These data can however be found in the supplementary data file should someone wish to use it for their own research.

Reviewer 3 Report

This manuscript provides new data on past environmental change from the understudied Central American Dry Corridor. My main comments concern the fossil diatom assemblages. The authors need to expand more on some aspects (detailed below) to avoid the perception that the diatom interpretation is forced to fit the Stansell et al 2013 d18O precipitation reconstruction. With minor-to-moderate revisions this paper will be suitable for publication.

Main comments

Line 362. Subject to wetter conditions, El Gancho’s basin is suggested to have expanded laterally, increasing in area but not necessarily depth.

This needs some additional explanation. With increased precipitation water levels would have to rise especially in the deepest portion of the basin where the core was recovered. Is the assumption here that the lake was already at its maximum depth? If so, this should be made explicit.

Line 369, 370. What are disturbance diatom taxa? This is a meaningless term does not define any group of diatoms. Should rephrase to diatoms indicative of changing conditions.

Line 371. The succession of Aulacoseira ambigua at 1340-1600C.E. is suggested to represent a rapid reduction in lake area, whereby the water occupying El Gancho retreated to the deepest part of its basin, abandoning the expansive shallow periphery.

OK, but with max depth of 1.1 m, would not all of the habitat (even in the deepest portion of the basin) be available habitat for benthic diatoms? Also, if the lake was previously deep (and large in area) during the wet MCA (950-1250) then why is A. ambigua largely absent? This would also have been suitable habitat for planktonic diatoms. In other words, why does A. ambigua only appear after the lake begins to shrink in area? Was it present before, but just diluted in numbers by all the benthic species. And if the area surrounding the deep basin has a shallow slope, is this enough for sediment focussing to transport benthic taxa from outlying areas to the center of the deep basin?   

Section 4.2

The pollen zones are explained by changes in wet vs dry conditions, yet the MCA spans both zones 1 and 2 and the LIA spans both zones 2 and 3. Why this offset? Is the timing of the MCA and LIA different here?

Line 422. In addition, select arboreal taxa reflect deforestation for building materials e.g. Bursera simaruba, and Protium

This is unclear. Do you mean a decline in abundance of Bursera and Protium pollen indicate deforestation? They do show slight declines since 1500 CE, but seem to be within the range of natural variability (?)

Line 426. Dung fungal spore evidence indicates that animal abundance on the Asese peninsula increased after the arrival of the Spanish through until 1650C.E.

OK, but Sporomiella was present in equally high abundances (higher?) from around 950 to 1200 CE. How is this explained then if the higher abundances post 1500s are attributed solely cattle. What caused the earlier increases in Sporomiella?

Line 482. Rapid agricultural growth from c.1800C.E. has allowed for larger populations to be supported; however, it has also increased the reliance on agriculture 

Is there no evidence for rapid agricultural growth in the pollen record?

Some minor comments

Fig. 1. Scale-bar for larger map needed.

Fig. 6. Text for axes labels, species etc could be larger. This applies to other figs as well.

Line 15. DTF, not DFT

Line 36. Given “this recent episode”. You only gave one example, not several.

Line 48. Is forcing’s correct? Forcings?

Line 51. Remove the a period after “and”

Line 55. Should this one sentence paragraph be combined with para at line 58?

Line 69. In comparison, not comparisons.

Line 71. Awkward wording, “reduced evidence for population”? Do you mean “evidence of reduced population”

Line 75. For example, a study by Dull [27]...

Line 77. found that anthropogenic burning between c.500B.C.E. – 1500C.E. benefited a more open grassland environment at the expense of woody taxa.

Better to replace “benefited” with “resulted in”?

Line 87. Similarly, DTF will either be (i) vulnerable to anthropogenic activities, such as agrarian practices, land conversion and population density because of land use change; or (ii) they will be resilient to anthropogenic factors due to composition adaptation over time [35].

This is unclear to me. How will DTF be resilient to deforestation and burning / land conversion by humans?

Line 108. Do not italicize “and”

Line 114. km2?

Line 118. Should “milpa” be italiziced? Capitalized?

Line 143. Closed basin lake

Line 177. Do you mean extracted at 4 cm intervals?

Line 235. Interpolation?

Line 267. Italicize Cyclotella meneghiniana

Line 277. This is first mention of d18O. Readers would benefit from a brief description of how it was interpreted in Stansell et al [16] and main findings.

Line 355. Remove “within”

Author Response

The apparent resilience of the dry tropical forests of the Nicaraguan region of the Central American dry corridor to variations in climate over the last c.1200 years

The authors would like to thank all three reviewers for their insightful and constructive comments for improving this manuscript and hope that they find the below responses and additions to the manuscript satisfactory.

Reviewer 3

This manuscript provides new data on past environmental change from the understudied Central American Dry Corridor. My main comments concern the fossil diatom assemblages. The authors need to expand more on some aspects (detailed below) to avoid the perception that the diatom interpretation is forced to fit the Stansell et al 2013 d18O precipitation reconstruction. With minor-to-moderate revisions this paper will be suitable for publication.

1.      Line 362. Subject to wetter conditions, El Gancho’s basin is suggested to have expanded laterally, increasing in area but not necessarily depth. This needs some additional explanation. With increased precipitation water levels would have to rise especially in the deepest portion of the basin where the core was recovered. Is the assumption here that the lake was already at its maximum depth? If so, this should be made explicit.

a.      The assumption is that El Gancho was already at its maximum depth – this sentence has been added into the main body of text.

2.      Line 369, 370. What are disturbance diatom taxa? This is a meaningless term does not define any group of diatoms. Should rephrase to diatoms indicative of changing conditions.

a.      Thank you for the suggestion, the terms “disturbance diatom taxa” have been replaced by “diatoms indicative of changing conditions”

3.      Line 371. The succession of Aulacoseira ambigua at 1340-1600C.E. is suggested to represent a rapid reduction in lake area, whereby the water occupying El Gancho retreated to the deepest part of its basin, abandoning the expansive shallow periphery. OK, but with max depth of 1.1 m, would not all of the habitat (even in the deepest portion of the basin) be available habitat for benthic diatoms? Also, if the lake was previously deep (and large in area) during the wet MCA (950-1250) then why is A. ambigua largely absent? This would also have been suitable habitat for planktonic diatoms. In other words, why does A. ambigua only appear after the lake begins to shrink in area? Was it present before, but just diluted in numbers by all the benthic species. And if the area surrounding the deep basin has a shallow slope, is this enough for sediment focussing to transport benthic taxa from outlying areas to the center of the deep basin?

a.      The whole lake habitat would be suitable for benthic diatoms to thrive if these taxa were not outcompeted by diatoms indicative of changing conditions i.e. the Aulacoseira spp. and Pseudostaurosira brevistriata. The rapid expansion and retraction of this water body between the wet season and the dry season is through to have facilitated continuous biannual disturbance promoting a habitat more conducive for these two taxa to dominate the assemblage.

b.      During the transition from the MCA, when we suggest that El Gancho began to retreat in area, the large shallow expanse surrounding the central depocenter would have been subject to the greatest variation in size. Given that Pseudostaurosira brevistriata is a facultatively planktonic taxa we suggest that the available habitat niche provided conditions amiable to this taxa’s success over other taxa present such as Aulacoseira ambigua. In effect other taxa were diluted as suggested.

c.      Even though the area surrounding El Gancho is shallow, we believe that the variation between seasons, causing the lake to expand and contract would be enough to transport diatom taxa from across the whole basin.

4.      The pollen zones are explained by changes in wet vs dry conditions, yet the MCA spans both zones 1 and 2 and the LIA spans both zones 2 and 3. Why this offset? Is the timing of the MCA and LIA different here?

a.      The MCA and LIA imposed on the diagrams here are taken from Stansell et al. (2013). It is possible that the biological proxies are reflecting a lag response. Please see the addition to the text.

5.      Line 422. In addition, select arboreal taxa reflect deforestation for building materials e.g. Bursera simaruba, and Protium. This is unclear. Do you mean a decline in abundance of Bursera and Protium pollen indicate deforestation? They do show slight declines since 1500 CE, but seem to be within the range of natural variability (?)

a.      Thank you for this comment, we have clarified this point in the text.

6.      Line 426. Dung fungal spore evidence indicates that animal abundance on the Asese peninsula increased after the arrival of the Spanish through until 1650C.E. OK, but Sporomiella was present in equally high abundances (higher?) from around 950 to 1200 CE. How is this explained then if the higher abundances post 1500s are attributed solely cattle. What caused the earlier increases in Sporomiella?

a.      Please see additional text incorporated to explain the abundance of Sporormiella prior to European contact.

7.      Line 482. Rapid agricultural growth from c.1800C.E. has allowed for larger populations to be supported; however, it has also increased the reliance on agriculture. Is there no evidence for rapid agricultural growth in the pollen record?

a.      We found no palynological evidence for agriculture in our record.

8.      Fig. 1. Scale-bar for larger map needed.

a.      Thank you, please see the new figure.

9.      Fig. 6. Text for axes labels, species etc could be larger. This applies to other figs as well.

a.      Thank you for this suggestion, please see larger text on Figure 6. Text cannot be made any larger in other figures without selectively removing some taxa and losing detail. Given the intricate details we hope the editor will allow for a full page in the journal for figures 4, 5, 6, & 7.

10.   Line 15. DTF, not DFT

a.      Thank you – correction implemented.

11.   Line 36. Given “this recent episode”. You only gave one example, not several.

a.      Thank you – correction implemented.

12.   Line 48. Is forcing’s correct? Forcings?

a.      Thank you – correction implemented.

13.   Line 51. Remove the a period after “and”

a.      Thank you – correction implemented.

14.   Line 55. Should this one sentence paragraph be combined with para at line 58?

a.      Thank you – correction implemented.

15.   Line 69. In comparison, not comparisons.

a.      Thank you – correction implemented.

16.   Line 71. Awkward wording, “reduced evidence for population”? Do you mean “evidence of reduced population”

a.      Thank you – correction implemented.

17.   Line 75. For example, a study by Dull [27]...

a.      Thank you for this suggestion – an additional start to this sentence has been included as suggested by Reviewer 1. “All three of these studies by Dull…”

18.   Line 77. found that anthropogenic burning between c.500B.C.E. – 1500C.E. benefited a more open grassland environment at the expense of woody taxa. Better to replace “benefited” with “resulted in”?

a.      Thank you – correction implemented.

19.   Line 87. Similarly, DTF will either be (i) vulnerable to anthropogenic activities, such as agrarian practices, land conversion and population density because of land use change; or (ii) they will be resilient to anthropogenic factors due to composition adaptation over time [35]. This is unclear to me. How will DTF be resilient to deforestation and burning / land conversion by humans?

a.      There will be persistence of DTF taxa (as indicated in the palynological abundance) when there is deforestation and burning / land conversion by humans.

20.   Line 108. Do not italicize “and”

a.      Thank you – correction implemented.

21.   Line 114. km2?

a.      Thank you – correction implemented.

22.   Line 118. Should “milpa” be italiziced? Capitalized?

a.      No. Thank you – correction implemented.

23.   Line 143. Closed basin lake

a.      Thank you – correction implemented.

24.   Line 177. Do you mean extracted at 4 cm intervals?

a.      Thank you – “intervals” added.

25.   Line 235. Interpolation?

a.      Thank you – correction implemented.

26.   Line 267. Italicize Cyclotella meneghiniana

a.      Thank you – correction implemented.

27.   Line 277. This is first mention of d18O. Readers would benefit from a brief description of how it was interpreted in Stansell et al [16] and main findings.

a.      Please see addition in the text. Line 282.

28.   Line 355. Remove “within”

a.      Thank you – correction implemented.

Round 2

Reviewer 1 Report

The authors have provided overall satisfactory explanations to most of my questions and comments. However, the interpretation of the diatom assemblages still needs further explanation; in its actual form it seems that the authors are trying to accommodate the diatom data to the dry/wet periods inferred from the oxygen isotopes. The diatom data is poorly explored and since it is only used as a support to the isotope data, I wonder if it is useful at all. The record is more valuable if interpreted in depth considering changes in macrophytes, pH and nutrients brought about by changes in precipitation/human impact. Without this discussion, again, it seems the authors are trying hard to match the isotope and diatom data.

This can be solved in part by including and expanding the explanations provided to Reviewer 3, Comment 3 responses a,b and c) on the succession of Pseudostaurosira, Aulacoseria and back to Pseudostaurosira in function of pond enlargement/contraction. Having said that, it is important that the authors: (1) consider the fact that the absence of Aulacoseira in Zone 1 could be due to competition with cyanobacteria as proposed by Whitmore et al. 2018 “Cyanobacterial influence on diatom community lifeform dynamics in shallow subtropical lakes of Florida USA, JOPL, and (2) use the ecology of Cyclotella meneghiniana in order to bring a deeper discussion to the diatom data. This species is mentioned frequently in the description of the zones, but not use at all in the interpretation. This species can be providing key limnological information on nutrients and salinity (see Fritz et al. 2019 JOPL).

Finally, the classification of the species Navicula longicephala and N. penssylvanica is in my opinion, incorrect.

Author Response

The authors would like to thank the reviewer for their continued suggestions for the improvement of this manuscript.

The authors have provided overall satisfactory explanations to most of my questions and comments. However, the interpretation of the diatom assemblages still needs further explanation; in its actual form it seems that the authors are trying to accommodate the diatom data to the dry/wet periods inferred from the oxygen isotopes. The diatom data is poorly explored and since it is only used as a support to the isotope data, I wonder if it is useful at all. The record is more valuable if interpreted in depth considering changes in macrophytes, pH and nutrients brought about by changes in precipitation/human impact. Without this discussion, again, it seems the authors are trying hard to match the isotope and diatom data.

1.      This can be solved in part by including and expanding the explanations provided to Reviewer 3, Comment 3 responses a,b and c) on the succession of Pseudostaurosira, Aulacoseria and back to Pseudostaurosira in function of pond enlargement/contraction.

a.      Integration of the responses to the comments from Reviewer 3 have been incorporated into the diatom discussion as suggested to further clarify and highlight how the diatoms have been used to interpret basin shape and depth – lines 379-389.

2.      Having said that, it is important that the authors: (1) consider the fact that the absence of Aulacoseira in Zone 1 could be due to competition with cyanobacteria as proposed by Whitmore et al. 2018 “Cyanobacterial influence on diatom community lifeform dynamics in shallow subtropical lakes of Florida USA, JOPL, and (2) use the ecology of Cyclotella meneghiniana in order to bring a deeper discussion to the diatom data. This species is mentioned frequently in the description of the zones, but not use at all in the interpretation. This species can be providing key limnological information on nutrients and salinity (see Fritz et al. 2019 JOPL).

a.      As responded to Reviewer 1 in the previous round of recommendations: “We previously explored the salinity and nutrient groupings for all diatom taxa and did not find conclusive trends associated with the most likely scenario presented in our manuscript.” We have added this sentence into the manuscript lines 395-396 to make it clear that we did consider this aspect.

b.     While additional discussion on the prevalence/absence of particular taxa or the potential impact of cyanobacterial interactions may be interesting, we believe that is outside the scope of this study and would detract from the overall interpretation of the collective diatom assemblage, weakening the narrative.

3.      Finally, the classification of the species Navicula longicephala and N. penssylvanica is in my opinion, incorrect.

a.      There are many different taxonomic systems used locally and globally. We have been consistent in our use and referenced it appropriately to the relevant taxonomic literature in the manuscript. We therefore stand by the classification that we use.

Reviewer 2 Report

Line 290: replace "native" with "negative"

Author Response

Line 290: replace "native" with "negative"

Thank you -  this correction has been implemented.